# The role of maternal preconception vitamin D status in human offspring sex ratio

Alexandra C. Purdue-Smithe [1], Keewan Kim[1], Carrie Nobles[1], Enrique F. Schisterman[1], Karen C. Schliep[2], Neil J. Perkins[3], Lindsey A. Sjaarda[1], Joshua R. Freeman[1], Sonia L. Robinson[1], Jeannie G. Radoc[1], James L. Mills[1], Robert M. Silver[2], Aijun Ye[1] & Sunni L. Mumford [1]✉

Evolutionary theory suggests that some animal species may experience shifts in their offspring sex ratio in response to maternal health and environmental conditions, and in some unfavorable conditions, females may be less likely to bear sons. Experimental data in both animals and humans indicate that maternal inflammation may disproportionately impact the viability of male conceptuses; however, it is unknown whether other factors associated with both pregnancy and inflammation, such as vitamin D status, are associated with the offspring sex ratio. Here, we show that among 1,228 women attempting pregnancy, preconception 25-hydroxyvitamin D concentrations are positively associated with the live birth of a male infant, with notably stronger associations among women with elevated high sensitivity C-reactive protein, a marker of systemic low-grade inflammation. Our findings suggest that vitamin D may mitigate maternal inflammation that would otherwise be detrimental to the implantation or survival of male conceptuses in utero.

[1] Epidemiology Branch, Division of Intramural Population Health Research, Eunice Kennedy Shriver National Institute of Child Health and Human Development, National Institutes of Health, Bethesda, MD, US. [2] Department of Obstetrics and Gynecology, University of Utah, Salt Lake City, UT, US. [3] Biostatistics and Bioinformatics Branch, Division of Intramural Population Health Research, Eunice Kennedy Shriver National Institute of Child Health and Human Development, National Institutes of Health, Bethesda, MD, US. ✉email: mumfords@mail.nih.gov

According to evolutionary theory, the ability of some animal species to adjust their offspring sex ratio in response to maternal health status and resource availability confers a natural selection advantage, and in some unfavorable circumstances, females may be less likely to bear sons[1]. Over the past several decades, a significant decline in the male to female human sex ratio has been observed in North America[2,3], Europe[4,5], and Asia[6]. Although factors influencing this shift are unclear, several lines of evidence suggest that maternal inflammation may be specifically harmful to the implantation or survival of male embryos[7,8], which could contribute to sex ratio reduction on the population scale. Indeed, we demonstrated previously in the Effects of Aspirin in Gestation and Reproduction [EAGeR] trial that preconception-initiated low-dose aspirin (LDA) administration restored a reduced sex ratio at birth among women with elevated inflammation, suggesting that this phenomenon may also occur in humans[9]. Exposure to smoking[10], methylmercury[11], and earthquakes[12] were similarly associated with a reduced sex ratio in observational studies, further indicating that the proportion of male to female live births in a given population may be sensitive to an array of environmental stressors, possibly through inflammatory pathways. Ecologic data also suggest seasonal, altitudinal, and latitudinal sex ratio gradients in some regions of the world, with reduced rates of male live birth at higher lines of latitude, lower altitude, and following periods of cold temperatures[13–17], implying a potential role of vitamin D, which is synthesized in the epidermis during periods of sun exposure.

Considering the detrimental effect of maternal inflammation on male implantation or survival, vitamin D may be beneficially related to offspring sex ratio through its immunomodulatory effects[18]. Vitamin D appears to regulate placental inflammation and may be involved in the immunologic response necessary for successful embryo implantation and survival[19,20]. In a recent prospective study, we reported that preconception 25-hydroxyvitamin D [25(OH)D] concentrations were positively associated with rates of clinical pregnancy and live birth, and inversely related to pregnancy loss in healthy women, indicating that adequate vitamin D may improve implantation and fetal survival, perhaps by modulating inflammation[21]. Preconception 25(OH)D concentrations were also positively associated with fecundability in two other prospective studies of healthy pregnancy planners[22,23], as well as in couples seeking fertility treatment[24,25]. However, to our knowledge, no prior studies have evaluated whether these observed associations vary according to fetal sex, translating to alteration of the offspring sex ratio.

The objective of our study was to evaluate associations of preconception 25(OH)D concentrations with male live birth among 1288 reproductive-age women attempting pregnancy in the Effects of Aspirin in Gestation and Reproduction (EAGeR) trial from 2007 to 2011. Here, we show that women with sufficient concentrations of 25(OH)D ( $\geq$ 30 ng mL$^{-1}$) are 26% (RR = 1.26; 95% CI: 1.03, 1.53) more likely to have a live-born male infant compared to those with insufficient concentrations (<30 ng mL$^{-1}$). These associations are stronger among women with elevated preconception high sensitivity C-reactive protein (hsCRP) (>1.95 ng mL$^{-1}$: RR = 1.44; 95% CI: 0.99, 2.11) versus low preconception hsCRP ($\leq$1.95 ng mL$^{-1}$ RR = 1.12; 95% CI: 0.89, 1.41), a marker of systemic low-grade inflammation. Our findings that preconception vitamin D status is associated with male live birth, particularly among women with elevated inflammation, suggest that sufficient levels of vitamin D may mitigate maternal inflammation that would otherwise be detrimental to the implantation or survival of male conceptuses in utero. These hypothesis-generating findings lend further evidence to the importance of vitamin D in reproduction and suggest a role of vitamin D in the offspring sex ratio in humans.

## Results

Among 1191 women with available preconception serum 25(OH) D, the overall 25(OH)D mean (SD) was 30.8 (12.2) ng mL$^{-1}$ (range = 5.0 to 143.6 ng mL$^{-1}$). Fifty three percent ($n$ = 636) of women were vitamin D insufficient and 47% ($n$ = 555) were sufficient (Table 1). On average, women with sufficient preconception vitamin D status were thinner and reported higher household income, levels of physical activity, and alcohol consumption than women with insufficient vitamin D status. Women with sufficient vitamin D were also more likely to be employed and non-Hispanic white than those with insufficient vitamin D.

The primary analysis aimed to evaluate the probability of carrying and giving birth to a live-born male associated with preconception vitamin D status. Among all 1094 (including 8 twin gestations) women who completed follow-up, each 10 ng mL$^{-1}$ increase in preconception 25(OH)D was associated with an 8% higher probability of a live-born male (unadjusted RR = 1.08; 95% CI = 1.00, 1.17) (Table 2). Likewise, women who had sufficient preconception 25(OH)D concentrations were more likely to give birth to a live-born male compared to those who had insufficient 25(OH)D concentrations (unadjusted RR = 1.27, 95% CI = 1.04, 1.56). This positive association persisted after adjusting for age, race/ethnicity, and number of previous live births in multivariable analyses (sufficient versus insufficient adjusted RR = 1.26, 95% CI = 1.03, 1.53). In absolute terms, vitamin D sufficiency compared to insufficiency was associated with 5.60 additional male live births per 100 women (adjusted RD per 100 women = 5.60; 95% CI = 0.35, 10.85) (Supplementary Table 1). Results from restricted cubic spline models supported the linear fit of models evaluating continuous 25(OH)D and male live birth (Fig. 1). Vitamin D status was not associated with probability of female live birth among women who completed follow-up (sufficient versus insufficient adjusted RR = 1.05, 95% CI = 0.87, 1.28) (Supplementary Table 2).

In analyses stratified by preconception hsCRP, associations between preconception vitamin D status and male live birth were notably stronger in women with elevated inflammation (Table 2). Among women with hsCRP > 1.95 ng mL$^{-1}$, the adjusted RR for male live birth comparing vitamin D sufficiency versus insufficiency was 1.44 (95% CI = 0.99, 2.11). Conversely, among women with hsCRP levels $\leq$1.95 ng mL$^{-1}$, vitamin D status was not strongly associated with probability of male live birth (sufficient versus insufficient adjusted RR = 1.12, 95% CI = 0.89, 1.41). The Spearman rank correlation coefficient between baseline 25(OH)D and hsCRP was $-0.19$ (95% CI = $-0.24$, $-0.13$).

Our pregnancy-weighted analyses ($n$ = 803 including 8 twins) yielded estimates similar to those calculated among the total cohort of women who completed follow-up, though slightly attenuated with wider confidence intervals (Table 2, Supplementary Table 1). For example, among women who became pregnant, the adjusted RR for male live birth comparing vitamin D sufficiency versus insufficiency was 1.18 (95% = 0.97, 1.44). Among women with hsCRP >1.95 ng mL$^{-1}$, those with sufficient compared to insufficient vitamin D status were 25% more likely to have a live-born male (95% CI = 0.85, 1.84). Among women with hsCRP $\leq$1.95 ng mL$^{-1}$, vitamin D status was not strongly associated with male live birth (sufficient versus insufficient adjusted RR = 1.06, 95% CI = 0.84, 1.32).

Similarly, our live birth-weighted analyses ($n$ = 601 including 6 twin gestations) produced estimates that were similar to those among all women with complete follow-up and among pregnancies; for example, the adjusted RR for male live birth comparing vitamin D sufficiency versus insufficiency was 1.18 (95% CI = 0.94, 1.48) (Table 2). Among women with hsCRP >1.95 ng mL$^{-1}$, those with sufficient compared to insufficient vitamin D status were 25% more likely to have a male live birth (95% CI = 0.81,

**Table 1 Study population characteristics by preconception vitamin D status in the Effects of Aspirin in Gestation and Reproduction (EAGeR) trial, 2007–2011[a,b].**

| | | 25(OH)D | |
|---|---|---|---|
| | Total | <30 ng mL$^{-1}$ | ≥30 ng mL$^{-1}$ |
| N | 1191 | 636 (53) | 555 (47) |
| Age, y | 28.7 (4.8) | 28.6 (4.8) | 28.9 (4.8) |
| BMI, kg m$^{-2}$ | 26.3 (6.6) | 27.8 (7.3) | 24.5 (5.1) |
| Waist–hip ratio | 0.81 (0.07) | 0.82 (0.08) | 0.79 (0.06) |
| Race/ethnicity | | | |
| White | 1128 (95) | 586 (92) | 542 (98) |
| Non-white | 63 (5) | 50 (8) | 13 (2) |
| Education | | | |
| > High school | 1033 (87) | 542 (85) | 491 (89) |
| ≤ High school | 158 (13) | 94 (15) | 64 (12) |
| Season of blood collection | | | |
| Winter | 268 (23) | 147 (23) | 121 (22) |
| Spring | 342 (29) | 192 (30) | 150 (27) |
| Summer | 270 (23) | 134 (21) | 136 (25) |
| Fall | 311 (26) | 163 (26) | 148 (27) |
| Vitamin use | | | |
| No | 86 (7) | 43 (7) | 43 (8) |
| Yes, with folic acid | 946 (81) | 517 (83) | 429 (78) |
| Yes, with no folic acid | 143 (12) | 64 (10) | 79 (14) |
| Smoking in the past year | | | |
| Never | 1033 (88) | 551 (87) | 482 (87) |
| <6 times/week | 85 (7) | 42 (7) | 43 (8) |
| Daily | 63 (5) | 37 (6) | 26 (5) |
| Household income | | | |
| ≥$100,000 | 470 (40) | 261 (41) | 209 (38) |
| $75,000–$99,999 | 147 (12) | 63 (10) | 84 (15) |
| $40,000–$74,999 | 174 (15) | 75 (12) | 99 (18) |
| $20,000–$39,999 | 307 (26) | 179 (28) | 128 (23) |
| ≤$19,999 | 92 (8) | 57 (9) | 35 (6) |
| Employed | | | |
| Yes | 871 (73) | 448 (70) | 423 (76) |
| No | 278 (23) | 158 (25) | 120 (22) |
| Missing | 42 (4) | 30 (5) | 12 (2) |
| Time from last loss to randomization | | | |
| ≤4 | 630 (54) | 325 (52) | 305 (56) |
| 5 to 8 | 215 (18) | 108 (17) | 107 (20) |
| 9 to 12 | 98 (8) | 59 (9) | 39 (7) |
| >12 | 229 (20) | 135 (22) | 94 (17) |
| Number of previous live births | | | |
| 0 | 550 (46) | 279 (44) | 271 (49) |
| 1 | 433 (36) | 231 (36) | 202 (36) |
| 2 | 208 (18) | 126 (20) | 82 (15) |
| Number of previous losses | | | |
| 1 | 799 (67) | 425 (67) | 374 (67) |
| 2 | 392 (33) | 211 (33) | 181 (33) |
| Eligibility strata | | | |
| Original | 531 (45) | 274 (43) | 257 (46) |
| Expanded | 660 (55) | 362 (57) | 298 (54) |
| Alcohol consumption in the past year | | | |
| Never | 782 (67) | 445 (71) | 337 (61) |
| Sometimes | 368 (31) | 168 (27) | 200 (36) |
| Often | 26 (2) | 14 (2) | 12 (2) |
| Physical activity | | | |
| Low | 310 (26) | 183 (29) | 127 (23) |
| Moderate | 491 (41) | 245 (39) | 246 (44) |
| High | 390 (33) | 208 (33) | 182 (33) |

[a]37 women missing 25(OH)D.
[b]Values are means (SD) or N (%).

1.92). Among women with hsCRP ≤1.95 ng mL$^{-1}$, vitamin D sufficiency versus insufficiency was not strongly associated with male live birth (adjusted RR = 1.02, 95% CI = 0.82, 1.27).

In exploratory analyses that evaluated the probability of pregnancy with a male, women with sufficient preconception 25(OH)D concentrations were more likely to have a pregnancy with a male fetus compared to those with insufficient concentrations (adjusted RR = 1.22, 95% CI = 1.01, 1.47) (Supplemental Table 3). Estimates for pregnancy with a male fetus were also stronger among women with hsCRP >1.95 ng mL$^{-1}$ (sufficient versus insufficient adjusted RR = 1.34, 95% CI = 0.93, 1.92) compared to women with hsCRP ≤1.95 ng mL$^{-1}$ (sufficient versus insufficient adjusted RR = 1.11, 95% CI = 0.90, 1.38). Vitamin D status was not associated with pregnancy with a female fetus (sufficient versus insufficient adjusted RR = 1.06, 95% CI = 0.89, 1.26), or within strata of hsCRP.

Vitamin D status was not associated with pregnancy loss of a male conceptus—overall or within strata of aneuploid or euploid losses—among the 56 clinical pregnancy losses with available karyotype data (Supplemental Table 4). In the sensitivity analyses that imputed missing fetal sex for 134 pregnancy losses, RRs for pregnancy with male offspring comparing the vitamin D sufficient versus insufficient groups ranged from 0.83 to 1.69.

Associations for vitamin D status and male live birth were essentially unchanged in analyses restricted to White women (sufficient versus insufficient adjusted RR = 1.26, 95% CI = 1.03, 1.54) and to singleton pregnancies (sufficient versus insufficient adjusted RR = 1.27, 95% CI = 1.04, 1.55). Estimates for male live birth comparing vitamin D sufficient versus insufficient women were also robust to further adjustment for season of blood draw, study site, multivitamin use, and waist-to-hip ratio both in overall (sufficient versus insufficient adjusted RR = 1.24, 95% CI = 1.02, 1.52) and in stratified analyses (among hsCRP >1.95 ng mL$^{-1}$, sufficient versus insufficient adjusted RR = 1.54, 95% CI = 1.05, 2.27; among hsCRP ≤1.95 ng mL$^{-1}$, sufficient versus insufficient adjusted RR = 1.10, 95% CI = 0.87, 1.39). In post-hoc analyses stratified by eligibility criteria, estimates describing the association of vitamin D status and male live birth were pronounced in the original stratum among women with a single recent loss (sufficient versus insufficient adjusted RR = 1.48; 95% CI = 1.12, 1.96) compared to the women in the expanded stratum with 1 or 2 prior losses at any time in the past and at any gestational age (sufficient versus insufficient adjusted RR = 1.06; 95% CI = 0.79, 1.41) (Supplemental Table 5).

## Discussion

Collectively, our findings indicate that among women attempting to conceive, vitamin D sufficiency versus insufficiency during the preconception period is associated with higher probability of carrying and giving birth to a live-born male infant. Consistent with the hypothesized mechanism involving disordered maternal inflammation, we observed somewhat stronger associations among women with elevated hsCRP, a marker of chronic low-grade inflammation, where the reduced male to female infant sex ratio among women with higher versus lower inflammation was corrected among those who were vitamin D sufficient. Overall, no associations were observed for vitamin D status and female live birth, suggesting that sufficient preconception levels of vitamin D may ameliorate an inflammatory process that is disproportionately harmful to male conceptuses, restoring the sex ratio of newborn infants.

**Table 2 Unadjusted and adjusted relative risks (RRs) and 95% confidence intervals (CIs) for 25-hydroxyvitamin D [25(OH)D] and male live birth Effects of Aspirin in Gestation and Reproduction (EAGeR) trial, 2007–2011[a,b].**

| | $N^e$ | Among all women (n = 1094) | | Among pregnancies (n = 803)[c] | | Among live births (n = 601)[d] | |
|---|---|---|---|---|---|---|---|
| | | Unadjusted RR (95% CI) | Adjusted[f] RR (95% CI) | Unadjusted RR (95% CI) | Adjusted[f] RR (95% CI) | Unadjusted RR (95% CI) | Adjusted[f] RR (95% CI) |
| **25(OH)D** | | | | | | | |
| per 10 ng mL$^{-1}$ | 292 | 1.08 (1.00, 1.17) | 1.07 (0.99, 1.16) | 1.06 (0.98, 1.14) | 1.05 (0.98, 1.13) | 1.04 (0.95, 1.13) | 1.03 (0.95, 1.12) |
| <30 ng mL$^{-1}$ (referent) | 135 | 1 | 1 | 1 | 1 | 1 | 1 |
| ≥30 ng mL$^{-1}$ | 157 | 1.27 (1.04, 1.56) | 1.26 (1.03, 1.53) | 1.20 (0.98, 1.46) | 1.18 (0.97, 1.44) | 1.19 (0.94, 1.51) | 1.18 (0.94, 1.48) |
| **Among hsCRP≤1.95 ng mL$^{-1}$** | | | | | | | |
| <30 ng mL$^{-1}$ (referent) | 92 | 1 | 1 | 1 | 1 | 1 | 1 |
| ≥30 ng mL$^{-1}$ | 118 | 1.16 (0.92, 1.47) | 1.12 (0.89, 1.41) | 1.09 (0.87, 1.37) | 1.06 (0.84, 1.32) | 1.03 (0.83, 1.29) | 1.02 (0.82, 1.27) |
| **Among hsCRP>1.95 ng mL$^{-1}$** | | | | | | | |
| <30 ng mL$^{-1}$ (referent) | 43 | 1 | 1 | 1 | 1 | 1 | 1 |
| ≥30 ng mL$^{-1}$ | 39 | 1.41 (0.95, 2.09) | 1.44 (0.99, 2.11) | 1.27 (0.84, 1.92) | 1.25 (0.85, 1.84) | 1.34 (0.84, 2.13) | 1.25 (0.81, 1.92) |

[a]8 twin gestations contributed two observations each to the analysis.
[b]RRs and 95% CIs calculated using multiply-imputed generalized estimating equations of log-binomial regression with robust standard errors and stabilized inverse-probability weights to account for loss to follow-up. Poisson models used in cases of model non-convergence.
[c]Models further weighted to account for selection of pregnancies.
[d]Models further weighted to account for selection of pregnancies and survival to live birth.
[e]Number of live-born males.
[f]Multivariable model adjusted for age (continuous), race/ethnicity (white or non-white), number of previous live births (0, 1, or ≥2).

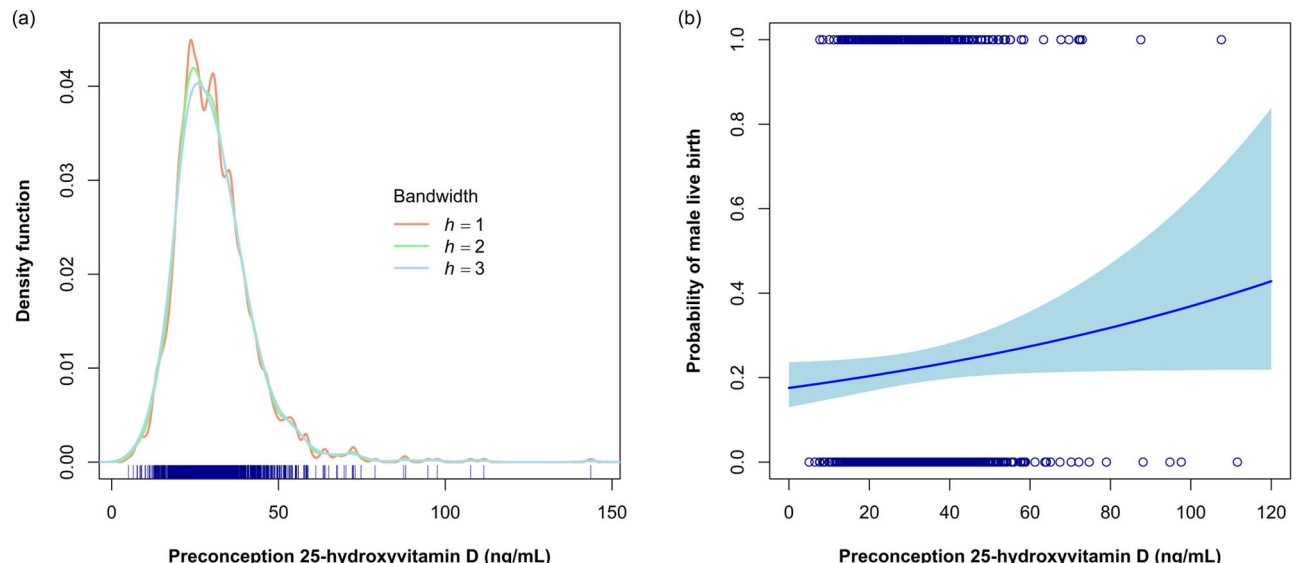

**Fig. 1 Distribution of preconception 25-hydroxyvitamin D [25(OH)D] levels and associations with male live birth in the Effects of Aspirin in Gestation and Reproduction (EAGeR) trial. a** Rug plot (navy) of preconception 25(OH)D levels with kernel density estimates for multiple bandwidth smoothers ($h = 1$, 2, and 3); **b** Scatterplot of vitamin D levels (navy) and probability of male live birth with regression line (blue) and 95% confidence cloud (light blue) in a single imputed dataset ($n = 1228$).

Although epidemiologic studies evaluating vitamin D and the offspring sex ratio are lacking, the notion that maternal nutrition and other factors may influence offspring sex ratio has notable precedent in the evolutionary biology literature. According to the Trivers–Willard sex allocation hypothesis, natural selection may favor species capable of shifting their offspring sex ratio in response to environmental stressors and maternal body condition in order to maximize the likelihood of passing on genetic material to grandchildren[1]. In some, but not all, mammalian species, males exhibit greater variance in terms of lifetime reproductive success, with much of this variance explained by specific advantageous physical and social characteristics[26]. For male offspring to

eventually join the select group exhibiting these characteristics, greater parental investment is theoretically required relative to female offspring; therefore, when conditions are poor, male offspring may be less likely than female offspring to eventually successfully reproduce. Conceptually, an underlying distribution of survivability of male embryos exists in any given population, and the threshold of maternal tolerance to those at the lower tail of this distribution apparently shifts downward with exposure to population stressors, selectively increasing the rate of loss among male embryos with lower survivability likelihood[27]. Though the consistency of studies of sex ratio varies widely, many studies report diminished sex ratios in human populations following

disasters[12], wars[28], and times of economic depression[29], and recent randomized clinical trial data implicate inflammation as a possible biologic pathway[9]. Less is known about the influence of maternal nutrition on the human sex ratio, but experimental data from ruminants, rodents, marsupials, and primates indicate that administration of low-calorie and low-fat diets, caffeine, and other dietary factors reduce the likelihood of male progeny[26]. Our results expand this body of research, suggesting that maternal nutrition—specifically, vitamin D status—may play a role in the offspring sex ratio of humans, possibly by "rescuing" male embryos that may have otherwise been lost in women with elevated inflammation.

The present findings also contribute to a growing body of evidence supporting a role of vitamin D in achieving and maintaining a healthy pregnancy. A recent analysis in the EAGeR cohort reported that women with sufficient preconception vitamin D had higher rates of pregnancy and live birth, as well as lower rates of pregnancy loss, relative to vitamin D insufficient women[21]. These findings are in agreement with two other prospective studies, which also reported lower fecundability and fertility among healthy women who were vitamin D insufficient or deficient ($25(OH)D < 12$ ng mL$^{-1}$) compared to those with sufficient concentrations[22,23]. Similar associations have been observed in clinical populations, in which $25(OH)D$ concentrations were positively associated with probability of implantation and pregnancy among infertile women undergoing IVF[24,25]. Importantly, these prior studies did not evaluate associations according to fetal sex, underscoring the research gap directly addressed by the present analysis. Indeed, our results extend these prior findings, demonstrating that vitamin D status may be specifically relevant to the probability of successfully establishing and/or maintaining a pregnancy with a male conceptus, more so than promoting pregnancy overall.

The exact mechanism through which vitamin D may be related to the offspring sex ratio is unclear and likely complex. Theoretically, aberration of the expected sex ratio of newborn infants must arise through differences in sex-specific probability of fertilization, implantation and/or survival in utero. In the context of previous data demonstrating the restoration of an altered sex ratio by LDA among women with elevated inflammation[9], we suspect that as an immunomodulator with anti-inflammatory properties[18,30], vitamin D may act analogously by mitigating maternal inflammation that is specifically detrimental to males during at least one of these reproductive stages. Moreover, associations were particularly observed among women with a single recent loss, who tended to have lower adiposity on average, which suggests that the influence of vitamin D on male live birth may be stronger among women with non-obesity-related inflammation. The consistency of the estimates for vitamin D status and male live birth in our weighted analyses among all women who completed follow-up, pregnancies, and live births, in conjunction with the estimates for vitamin D and pregnancy with a male conceptus, suggest that the influence of preconception vitamin D status on the eventual live birth of a male begins at least as early as implantation, but does not necessarily rule out the potential for vitamin D to also act on male embryos after implantation via pregnancy loss.

In terms of sex-specific probability of implantation, bovine and ovine preimplantation embryos exhibit sexually dimorphic secretion of interferon-τ, the cytokine necessary for uterine receptivity signaling to sustain implantation[31], with female bovine preimplantation embryos producing nearly twice as much interferon-tau as their male counterparts[32]. In humans, hCG secretion by preimplantation embryos similarly contributes to maternal immunotolerance required for successful implantation[33], and although data on sex differences of hCG

production by human preimplantation embryos are unavailable, some evidence suggests small differences in hCG concentrations in pregnancies with surviving male versus female embryos during the week following implantation[34]. Though the precise mechanism is unclear, it is possible that lower hCG secretion by male preimplantation embryos may confer higher susceptibility to preimplantation failure among women with elevated inflammation, and adequate $25(OH)D$ levels may provide some remediation, thereby increasing the likelihood of survival and eventual implantation of male embryos. Our observed stronger positive associations for vitamin D status and pregnancy with a male fetus among women with elevated inflammation lend support to this possibility, though other physiologic mechanisms, such as those involving sex hormone levels are possible[35].

It is also possible that our findings may be partially explained by an influence of vitamin D on sexually dimorphic placental immune function involved in pregnancy maintenance after implantation. Several lines of evidence indicate that male placentae are more susceptible to maternal inflammation than female placentae[36]. For example, in human placental trophoblast cells, lipopolysaccharide stimulation increased overall output of TNF-α and IL-10, with significantly higher output by cells of male fetuses[37]. In another study, treatment of vitamin $D_3$ (calcitriol) prevented TNF-α-induced expression of proinflammatory cytokines in a dose-dependent manner in human trophoblasts[38]. In line with these experimental data, epidemiologic studies suggest that male fetuses are more susceptible to obstetric complications mediated by maternal inflammation, including preterm birth and placental infection[39]. Serum $25(OH)D$ concentrations, though not vitamin D supplementation[40], have been associated with a lower overall risk of these conditions[41], and some also reported effect modification by fetal sex, with stronger benefits among males[42]. Accordingly, our observed positive associations for vitamin D status and male live birth support a mechanism in which adequate vitamin D modulates inflammation that may otherwise be harmful to male survival in utero.

Our study has several limitations. First, we were primarily interested in the usual rather than acute (i.e., conception cycle) effects of vitamin D on male live birth, as our study did not include multiple measurements of preconception $25(OH)D$ with male live birth. Changes in $25(OH)D$ concentrations over time between blood draw and conception (~2 weeks to 5.5 months) may have resulted in some degree of misclassification of vitamin D status, which could be non-differential or differential. Reassuringly, 75% of women in the EAGeR trial conceived in the first 3 months of follow-up and data from a prior study of 68 healthy reproductive-age women suggest that $25(OH)D$ concentrations measured in the follicular phase of two consecutive cycles did not meaningfully change from cycle 1 to 2[43]. Moreover, in sensitivity analyses limited to conceptions occurring in the first 2 cycles of follow-up (~2 to 6 weeks after blood draw), estimates for male live birth comparing vitamin D sufficiency versus insufficiency were consistent with the overall analysis (among 430 pregnancies, adjusted RR = 1.24, 95% CI = 0.97, 1.59). Second, women in this study population were relatively vitamin D replete, which limited power to evaluate more extreme cut-points for vitamin D and offspring sex ratio. Third, as in any observational study, it is possible that our observed associations may be influenced by unmeasured confounding, such as calcium or other dietary factors. Further, selection bias is possible if loss to follow-up or missingness is related to both vitamin D status and probability of pregnancy with a male or male live birth. Importantly, drop-out was low overall in the EAGeR cohort (11%), and we further addressed this possibility by weighting all analyses for loss to follow-up using stabilized inverse-probability weights. Importantly, the regression models used to generate inverse-probability

weights to account for selection bias arising from loss to follow-up and in analyses limited to pregnancies and live birth are subject to potential model misspecification. Necessary conditions of correct model specification require that the mean of the weights approximately equal to 1 and have no extreme outliers. On these bases, the distributions of our estimated weights support correct model specification; however, we acknowledge that complete elimination of selection bias is unlikely, as unknown or unmeasured common causes of selection factors (i.e., loss to follow-up, pregnancy, and live birth) and male live birth may exist.

Strengths of our study include a large preconception cohort and prospectively collected data and biomarkers. The EAGeR trial is an ideal population to study factors associated with altered sex ratio, as eligibility was limited to women with 1–2 prior pregnancy losses. Assuming that a biased sex ratio arises through sex-specific probabilities of implantation or survival, the elevated risk of loss and therefore, biased sex ratio, in our study population lends itself to efficiency in a prospective design. The trial also excluded women with conditions related to pregnancy loss, including polycystic ovary syndrome and pelvic inflammatory disease, reducing potential for confounding by these conditions, which may also be related to vitamin D status and inflammation. Further, among live births, the percentage of males in the vitamin D sufficient group (51%) translates to a sex ratio of 1.04 and matches the expected sex ratio at birth in the general population. We anticipate that our findings may therefore be generalizable to healthy women planning a pregnancy who have comparable levels of inflammation and similar reproductive history.

Our findings, in the context of available data, are the first to suggest that vitamin D may mitigate maternal inflammatory dysregulation that would otherwise be detrimental to the implantation of male conceptuses or their survival in utero, thus translating to altered sex ratio. Importantly, these data should not be interpreted as a method to influence infant sex, given that our results imply that vitamin D sufficiency restores the reduced probability of male live birth among women with elevated inflammation and would not impact the probability of spontaneously conceiving a male or female per se. Regardless, these findings shed light on the influence of maternal nutrition on offspring sex ratio in humans and lend additional evidence to the importance of the health status of women attempting to conceive, for whom vitamin D and inflammation status could impact the chances of establishing and maintaining a pregnancy after producing a male conceptus. Replication of these hypothesis-generating findings, as well as evaluations of other dietary and lifestyle factors that may influence sex ratio are warranted in future investigations.

## Methods

**Study population**. This was a secondary analysis of the Effects of Aspirin in Gestation and Reproduction (EAGeR) trial, a multi-center, block-randomized, double-blind, placebo-controlled clinical trial designed to evaluate the effect of preconception-initiated daily low-dose aspirin on reproductive outcomes in women with history of pregnancy loss. The trial design and overall results have been described elsewhere[44,45].

In brief, study participants included 1228 women attempting pregnancy, ages 18–40 years, with 1 to 2 prior pregnancy losses. Women with known history of infertility treatment, pelvic inflammatory disease, tubal occlusion, endometriosis, anovulation and polycystic ovary syndrome, or uterine abnormality were ineligible to participate. Participants were recruited at four clinical sites in the United States from 2007 to 2011 and followed for up to six menstrual cycles while attempting pregnancy and then throughout pregnancy for women who conceived. Among 1228 women randomized, 1088 completed follow up (89%).

Baseline demographic characteristics and reproductive history were obtained via questionnaire. Study-provided urine hCG pregnancy tests and fertility monitors assisted visit scheduling and were used to time intercourse (Clearblue Easy Fertility Monitor; Inverness Medical Innovations, Waltham, MA, USA). Women who had a positive pregnancy test visited the clinic at 6–7 weeks gestation to confirm a clinical

pregnancy, as defined by a visible gestational sac upon ultrasound. Pregnancies without a visible gestational sac at 6–7 weeks were classified as hCG pregnancy losses, and these women were instructed to continue pre-pregnancy follow-up through the remainder of the six months. Pregnancies that ended after ultrasound confirmation were considered clinical pregnancy losses. Ectopic pregnancies were defined as pregnancies that implanted outside of the uterus, as visualized by sonography, laparoscopy, or laparotomy, and were considered to be clinical pregnancy losses. Live birth was defined as a living infant born after 23 weeks gestation. There were no elective terminations during follow-up.

The institutional review board at each study site (Salt Lake City, Utah; Denver, Colorado; Buffalo, New York; Scranton, Pennsylvania) and data coordinating center approved the trial protocol and all participants provided written informed consent prior to enrolling. The trial was registered with ClinicalTrials.gov, number NCT00467363. This research complied with all relevant ethical regulations for work with human participants.

**Blood collection**. Preconception serum samples were collected prior to randomization and stored at −80 °C until analysis. 25(OH)D concentrations were measured using an ELISA solid-phase enzyme immunoassay (BioVendor R&D, Asheville, NC, USA), which is equipotent for $D_2$ and $D_3$, and is non-cross-reactive with 3-epi-25-hydroxyvitamin $D_3$. High-sensitivity C-reactive protein (hsCRP) was measured via immunoturbidimetric assay by a Roche COBAS 6000 autoanalyzer (Roche Diagnostics, Indianapolis, IN, USA).

For 25(OH)D, the mean intra-assay coefficients of variation were 8.2% at a concentration of 15.5 ng mL$^{-1}$ and 5.5% at a concentration of 41.6 ng mL$^{-1}$ for two manufacturer-lyophilized controls, and 5.6% at a concentration of 40.3 ng mL$^{-1}$ for the pooled control. The inter-assay laboratory coefficients of variation were 15.8% and 13.1% at mean concentrations of 15.5 and 41.6 ng mL$^{-1}$, respectively, for lyophilized manufacturer's controls, and 17% for the pooled serum control. All values were above the limit of detection (LOD) of 1.6 ng mL$^{-1}$. For hsCRP, inter-assay coefficients of variation were 5.1% at 1.05 ng mL$^{-1}$ and 6.7% at 3.12 ng mL$^{-1}$. There were 21 values of hsCRP below the LOD of 0.15 ng mL$^{-1}$. Values below the LOD were imputed as LOD/$\sqrt{2}$[46].

**Outcome assessment**. The biological sex of live-born infants was determined by medical chart abstraction. Among 603 live births (including 6 twin gestations), 2 live-born infants did not have available data on fetal sex. In the event of a clinical pregnancy loss, participants were asked to notify study personnel and to collect the products of conception using study-issued materials, where possible. The fetal/placental tissue was refrigerated for up to 24 h and then frozen at −80 °C until karyotype or chromosomal microarray could be performed. Among the 135 (including 2 twin gestations) clinical pregnancy losses, 84 underwent genetic testing. 55 tests determined fetal sex, whereas 29 were not determined due to testing failure. No genetic analysis was available for one phenotypic male (gestational age = 15 weeks). The median gestational age of karyotyped losses was 8 weeks (IQR: 7–10 weeks, range 2–20 weeks) (Fig. 2).

**Statistical analysis**. 1191 women had available preconception measures of 25(OH)D. Women were classified a priori as vitamin D insufficient (25(OH)D < 30 ng mL$^{-1}$) or sufficient (25(OH)D ≥ 30 ng mL$^{-1}$) according to Endocrine Society cut-points for optimal bone health, as cut-points for reproductive health have yet to be established[47]. We compared demographic and reproductive history characteristics according to baseline preconception vitamin D status using Student's $t$ tests and $\chi^2$ tests for continuous and categorical data, respectively. Multiple imputation using the fully conditional specification method[48] was used to address missing exposure and covariate data and all regression models accounted for imputation. Out of 1288 women who enrolled in the study, percentages of missing data ranged from $N = 1$ (0.1%) for physical activity to $N = 44$ (3.6%) for employment status for all predictor variables. Karyotype/fetal sex data was missing for $N = 134$ of 190 (71%) pregnancy losses and $N = 2$ of 603 (0.003%) live births.

An analysis to evaluate preconception vitamin D and sex ratio would ideally model the outcome at multiple stages of the reproductive process to determine whether vitamin D may influence the fertilization or implantation of male embryos, their survival in utero, or a combination of these events. Our statistical analyses aimed to approximate this scenario by estimating associations of vitamin D status with (1) probability of carrying and giving birth to a live-born male; and, (2) probability of pregnancy with a male embryo that implanted and was detected by urine hCG test, regardless of whether it survived to birth.

The primary analysis evaluated associations of vitamin D status with the probability of carrying and giving birth to a live-born male among all women who completed follow-up. Risk calculation using this denominator represents the population effect of preconception vitamin D status on male live birth in a real-world setting where among women attempting to conceive, some women go on to achieve pregnancy, while others may experience a pregnancy loss or not achieve pregnancy.

We estimated unadjusted and adjusted relative risks (RRs), risk differences (RDs) per 100 participants, and 95% confidence intervals (CIs) for male live birth according to continuous 25(OH)D and categorical vitamin D status using generalized estimating equations of log-binomial regression with robust standard

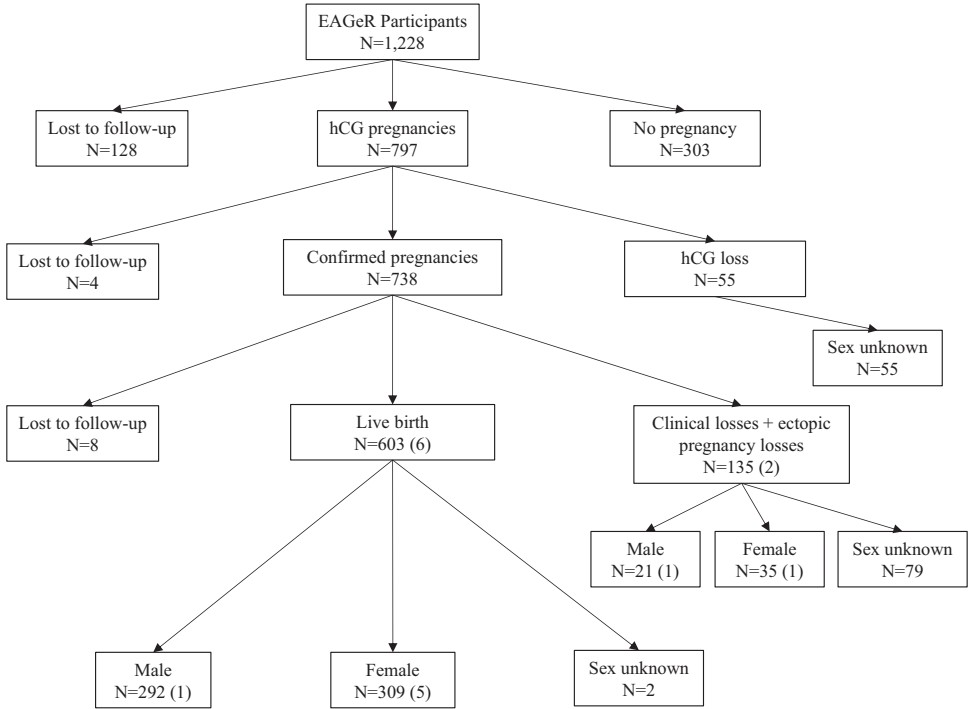

**Fig. 2 Flowchart of 1228 Effect of Aspirin in Gestation and Reproduction (EAGeR) trial participants.** Twin gestations are indicated in parentheses. The unit of analysis was the mother–offspring pair. 738 "confirmed pregnancies" include 732 clinical pregnancies + 6 non-viable ectopic pregnancies, which were considered to be clinical pregnancy losses upon ultrasound visualization.

errors to account for correlated data within the 8 twin gestations. We examined the possibly nonlinear relationship between continuous 25(OH)D and male live birth non-parametrically using restricted cubic spline models (with 3 to 5 knots specified) and evaluated the individual spline term contributions to the model fit and overall test for nonlinearity. Potential confounders were initially identified from prior literature on pregnancy and pregnancy loss[21,49] and final multivariable model specification was informed by directed acyclic graphs[50] (i.e., age, race/ethnicity, and number of previous live births). All models included stabilized inverse-probability weights to account for withdrawal or loss to follow-up during the study period, which were generated using the following factors: 25(OH)D, age, BMI, number of previous pregnancy losses, number of previous live births, LDA treatment group, marital status, and season of blood draw[51,52].

Because the primary analysis modeled male live birth among all women who completed follow-up, some of whom did not become pregnant, we repeated analyses among women with a positive urine hCG test during follow-up ($n = 803$ including 8 twin gestations). As vitamin D status may influence probability of pregnancy itself, we addressed potential selection bias introduced by conditioning on pregnancy using stabilized inverse-probability weights that accounted for the conditional probability of both complete follow-up and hCG-detected pregnancy[52]. The estimates from these pregnancy-weighted analyses are interpreted as the association of preconception vitamin D status with male live birth that would have been observed if all women who enrolled in EAGeR had completed follow-up and had an hCG-detected pregnancy. Variables used to generate the weights included factors associated with pregnancy in this cohort (i.e., 25(OH)D, age, BMI, number of previous pregnancy losses, number of previous live births, LDA treatment group, marital status, and season of blood draw).

We also conducted these analyses among live births ($n = 601$ including 6 twin gestations) and employed inverse-probability weights to account for the conditional probability of completed follow-up, pregnancy, and live birth. Predictors used to generate the weights were the same as those used to generate the pregnancy weights. The live birth-weighted estimates from these analyses describe the association of preconception vitamin D status with male live birth that would have been observed if all women in the study had completed follow-up and had an hCG-detected pregnancy that survived to birth. By limiting the denominator to live births and using inverse-probability weights to account for selection bias imposed by this restriction, we remove the influences of subfertility (not becoming pregnant) and pregnancy loss, and therefore estimate the biological effect of preconception vitamin D status on male live birth.

To address the a priori hypothesis of an association between vitamin D and sex ratio that may be modified by inflammation, we stratified all analyses according to preconception hsCRP (>1.95 versus ≤1.95 ng mL$^{-1}$). This cut-point corresponds to

the highest tertile of hsCRP in the study population, after excluding women with values >10 ng mL$^{-1}$, which may be indicative of acute inflammation resulting from injury or illness[53,54].

In exploratory analyses, we evaluated associations of vitamin D status and pregnancy with a male fetus utilizing the available karyotype data for 56 out of the 135 clinical pregnancy losses (including 2 twin gestations). To do so, we repeated analyses modeling pregnancy with a male fetus as the outcome among women who had a pregnancy with fetal sex determined and those who did not become pregnant. Pregnancy with a male fetus was defined as a male live birth or a pregnancy loss with a fetus identified as male from chromosomal analysis. We also compared proportions of male and female pregnancy losses in vitamin D sufficient versus insufficient groups, both overall and within strata of euploid/aneuploid status, using $\chi^2$ tests and Fisher's exact tests, respectively.

We conducted several sensitivity analyses to evaluate potential sources of bias and the degree to which they may have influenced the observed estimates. First, we repeated all analyses modeling female live birth and pregnancy with a female fetus as outcomes to determine whether the observed associations may be explained by excess female fetuses or live births. Second, to evaluate whether associations varied according to multiple gestation and race/ethnicity, we conducted analyses restricted to singleton pregnancies and to White women. Third, maternal cell contamination (MCC) may have resulted in misclassification of fetal sex of the pregnancy losses with available karyotype data. To determine the degree to which MCC may have influenced the observed estimates for pregnancy loss, we performed a simple quantitative bias analysis in which we corrected the proportions of male and female embryos according to findings of a prior validation study[55]. Finally, missing fetal sex of 134 pregnancy losses (79 clinical and 55 hCG-detected losses) and 2 live births may have introduced bias if missingness itself was related to both vitamin D status and fetal sex. To address this possibility, we imputed fetal sex for these pregnancy losses under every possible percentage male (0–100%) in the vitamin D insufficient and sufficient groups, and estimated RRs of the association between vitamin D status and male offspring for each hypothetical scenario. All analyses were conducted using SAS v9.4 software (Cary, NC) and R-4.0.3.

**Reporting summary**. Further information on research design is available in the Nature Research Reporting Summary linked to this article.

## Data availability

The data that support the findings of this study are available from the corresponding author upon reasonable request.

## Code availability

The code used to generate the results that appear in this manuscript can be found on GitHub (https://github.com/apurduesmithe/vitdsexratio).

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

## Acknowledgements

This research was supported by the Intramural Research Program of the *Eunice Kennedy Shriver* National Institute of Child Health and Human Development (National Institutes of Health, Bethesda, MD, USA; contract numbers HHSN267200603423, HHSN267200603424, and HHSN267200603426). Jeannie G. Radoc was supported by the NIH Medical Research Scholars Program, a public–private partnership supported jointly by the NIH and generous contributions

to the Foundation for the NIH from the Doris Duke Charitable Foundation (DDCF Grant # 2014194), Genentech, Elsevier, and other private donors.

## Author contributions

A.C.P.-S. and S.L.M. wrote the statistical analysis plan, cleaned and analyzed the data, and drafted and revised the paper. A.C.P.-S. and S.L.M. take full responsibility for the accuracy of the data presented. K.K., N.J.P., A.Y., and E.F.S. assisted with statistical analysis plan. K.K., C.N., K.C.S., L.A.S., E.F.S., J.R.F., S.L.R., J.G.R., J.L.M., and R.M.S. revised the draft paper. E.F.S. designed and implemented the trial.

## Competing interests

The authors declare no competing interests.

## Additional information

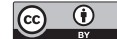

