## [Peer Review File · Nature Communications]

Reviewers' comments:

Reviewer #1 (Remarks to the Author):

"The role of maternal preconception vitamin D status in human offspring sex ratio"

This manuscript examines vitamin D status measured a varying amount of time prior to conception with offspring sex ratio measured both in pregnancies that are eventually lost and pregnancies that are delivered as live births. The results suggest that male conception is more likely in mothers who have sufficient vitamin D and that this association is stronger among women with higher levels of inflammation (measured by high sensitivity CRP). A strength of the paper is that the authors attempt to answer their research question with several analytical models, which supports the robustness of their results. Other comments meant to clarify the methods and results are presented below.

1. It appears that this analysis has only included linear and dichotomous parameterizations of 25(OH)D. It would be interesting to see a more flexible modeling strategy to determine the shape of the association, i.e. with splines, or multiple categories, or a quadratic function, etc.
2. I wonder if the paper should exclude twins? Twins are detectable early in gestation and presumably the controls and influences on sex ratio in monozygotic or dizygotic twins could be different than the influences on a singleton pregnancy. Do the authors have evidence that vitamin D would have the same influence on the sex ratio of twin gestations as singleton gestations?
3. Vitamin D is measured at enrollment, prior to randomization, and then women try to conceive for a certain amount of time and then if they conceive their pregnancies are examined for sex of the offspring. Given that 25(OH)D may change over time, and that one proposed mechanism of action is uterine preparation and implantation, I wondered if there is misclassification of the exposure, possibly differential? Perhaps I've missed it, but did the authors consider some form of imputation that would estimate the 25(OH)D level at conception rather than using the baseline measure alone?
4. Why are both risk ratios and risk differences presented? The results section doesn't seem to distinguish their interpretation and I don't think having both provides additional information, while it does make the tables more crowded. They could be moved to a supplement if the authors wish to keep them, and if they prefer them in the main paper, they should add some rationale or interpretation for each set of estimates.
5. Each study site probably varies in altitude, diet, air pollution, culture, climate, etc. Was study site related to sex ratio? And could it be a confounder of this analysis (or a surrogate for these underlying unmeasured factors)?
6. Relatedly, neither season nor vitamin use was associated with 25(OH)D in this population. Do the authors have any comment on this?
7. Induced abortion was likely rare in this population since they were trying to become pregnant, but can the authors confirm this in the methods?
8. What is the rationale for the variable "Number of previous pregnancies, not including losses"? This does not correspond to clinical gravidity and it is not clear, given the women in this study had a varying number of previous losses, why the losses are not considered conceptions?
9. Please describe if there were any values lower than the limit of detection for hsCRP. Also, add to the results the range of hsCRP values detected in the analysis sample.
10. "Risk calculation using this denominator reflects the true natural process in a given population of women attempting to conceive because all women are "at risk" of having a live-born male at the beginning of follow-up, though some will experience a pregnancy loss or may not achieve a pregnancy at all." How can a non-pregnant woman be "at risk" of having a male birth? If she is infertile, she can't be at risk of any birth. I think I understand the concept that women who appear to have not conceived may have actually conceived and had an undetectable loss, and perhaps that happens more often for male conceptuses, but I don't quite understand how this can be estimated in these data since none of that is observed?
11. Are the results of the primary analysis changed if non-White women are excluded from the analysis? I understand that race was an adjustment factor in the model, but this may not fully account for differences by race.
12. Figure 1 is difficult to follow in terms of whether it is by participant or by gestations. For example, there are $N=732$ confirmed pregnancies, but in the boxes below $603+135+8$ does not equal the above total of 732. Even if I account for twins, if my math is correct, $597+133+8$ does not add up to 732. So, I'm having trouble following the Ns through this figure. Perhaps check the math throughout and label the

units on each N to clarify whether it is participants or gestations in each box.

13. Introduction (lines 59-62): "Indeed, a large randomized placebo-controlled trial recently demonstrated that preconception-initiated low-dose aspirin administration restored a reduced sex ratio at birth among women with increased inflammation, providing direct evidence of this phenomenon in humans." Is this the same study that is analyzed in this paper? If so, it should be clarified that the human evidence for inflammation influencing the sex ratio is limited to this one cohort.

14. "Vitamin D appears to regulate placental inflammation and may be involved in the maternal-fetal immunologic response necessary for fetal survival.¹⁹" Is reference 19 the correct reference? It seems to say that vitamin D may be important for cellular differentiation (particularly in the uterus) with evidence from human cell lines. I don't immediately see a connection between that and "placental inflammation"? Could you explain further?

15. Does the lab that measured the 25(OH)D participate in DEQAS or the vitamin D standardization program?

16. What is the correlation between 25(OH)D and hsCRP?

17. I would find it helpful if the Discussion would explain why male conception is preferred when the mother is "healthy"? And how would inflammation selectively target male conceptuses?

Minor comments

Page 11, line 282, I believe "Ashville" is spelled incorrectly.

Tables 2 and 4 have EAGeR dates as 2007-2012, Table 1 has 2007-2011, and the text (line 260) had 2007-2011.

Table 2 has column header "Model 1," but this could be replaced with the header "Unadjusted" which is more informative. Similarly, "Model 2" and "Model 3" could be replaced with "Adjusted" with their corresponding footnotes to distinguish the adjustment sets.

Did any women enroll in EAGeR twice? Did any women have two pregnancy attempts in the study?

In Table 2 the N's aren't totally clear. The figure shows 292 male births total, but in Table 2 adding all of the male births in the hsCRP analysis, for example, results in 293 male births (88+111+48+46). Is that a typo?

Page 8, line 185, "LDA" is used here, although it was not defined as an acronym when the phrase was first used in the introduction.

Reviewer #2 (Remarks to the Author):

Review NCOMMS-19-11592: The role of maternal preconception vitamin D status in human offspring sex ratio.

The authors conducted a secondary analysis of data from the EAGeR study, a trial of pre-conception initiated low-dose aspirin treatment to investigate effects on live birth rate. The initial eligibility criteria required that participants had a single prior pregnancy loss (occurring at less than 20 weeks gestation and within the last year). A later, expanded eligibility criteria allowed for enrollment of women with one or two prior losses (loss could occur at any gestational age and anytime in past). This submitted manuscript addressed the question of whether preconception 25(OH)D concentrations were associated with sex ratio of the offspring (live births). The authors report that women with preconception 25(OH)D ≥ 30 ng/ml were more likely than the comparison group (women with lower 25(OH)D) to have boy babies. The association was stronger in women with C-reactive protein > 1.95 ng/ml (a marker of chronic inflammation).

This is a novel finding, and sex-ratio data tend to be of general interest. Unfortunately, sex-ratio findings rarely replicate.

The analytic approach was to start with the preconception cohort and use inverse probability weighting (IPW) to account for loss to follow-up, for the large subgroup that did not conceive an identifiable pregnancy, and for pregnancy losses before live birth. This is not the traditional approach to analyzing

sex-ratio data. The author's approach relies on being able to predict those transitions adequately with available data to assign appropriate weights. It is not clear how well that can be done. More importantly, the benefits of doing this are unclear. While it provides results that may be more representative of the study participant sample, the participants are volunteers with prior pregnancy losses, recruited under two different eligibility criteria. Thus, the external population to whom this is generalizable is uncertain. Also, the transition from trying to become pregnant to being pregnant and having a live birth is a fundamental state change. The sex of the baby is not a property of the woman until she is pregnant and can identify the sex. So, it seems inappropriate to start with the full study sample.

Additional major concerns.

1. 25(OH)D was measured from samples collected prior to randomization, but women took from 1 to 6 cycles to conceive. Thus, for some, the conception occurred in a different season than blood collection, and 25(OH)D is seasonal, especially in higher latitudes. The likelihood of misclassification of 25(OH)D due to a single measurement will increase with time to pregnancy. All of the women who did not conceive would likely have different concentrations by the end of their study attempts than the measured values.
2. The data on sex of pregnancy losses is highly selected and unlikely to be representative of all losses. Only a minority of the losses could be sexed, and whether or not they can be sexed is highly dependent upon gestational age at loss, sample collection, and probably the time between embryonic/fetal death and sample collection. The authors talk about sex ratio at implantation, but this is unobservable. Therefore, it is difficult to agree that their data "suggest that vitamin C sufficiency versus insufficiency is positively associated with survival and implantation of a male embryo particularly among women with higher inflammation."
3. In the initial EAGeR trial results paper, the two recruitment methods yielded somewhat different results. Was that taken into account in this secondary analysis?
4. The reported interaction between 25(OH)D and CRP has associated p-values of greater than 0.23, with the exception of one which was 0.14, so this interaction, though interesting, is not a strongly supported finding.

Specific comments/questions

1. Number of prior losses should be included in Table 1, as well as recruitment criteria.
2. The cut-point of 1.95 for CRP was at the highest tertile value. I am under the impression that 3.0 is often used in CVD risk. Is 1.95 a conceptually good cut-point? Also, were there women with such high levels of CRP that it is likely due to an acute health problem, not chronic inflammation? If so, how were these values dealt with?
3. Exactly how were twins entered into the analysis?
4. Why was waste/hip ratio used instead of BMI as a covariate?
5. The paragraph (lines 188-201) supporting male vs female differences in maternal recognition of pregnancy refer to non-human species. Mechanisms of maternal recognition of pregnancy vary dramatically within mammals. In humans, hCG is of primary importance. The only relevant data of which I am aware show no differences in the hCG rise during the week following implantation by sex at birth (Nepomnaschy et al., Human Reproduction, 2008).
6. Lines 218-219: Over the last 50 or more years there have been numerous publications reporting aberrant sex ratios associated with various factors, but consistent findings across several studies are generally lacking.
7. Inter-assay CVs are given, but not intra-assay CVs. The 25(OH)D CV for the blinded control was 17% which is quite high.
8. It would be helpful to list in a table the factors used in the IPW designed to account for each of the types of 'missingness' (able to conceive, conception survived to live birth, traditional loss to follow-up along the way, any others?). It was not easy to determine which factors were used for what.
9. The low-dose aspirin treatment of the trial (which can lower CRP) was treated as a weighting factor in IPW analyses. Was that done for all analyses? Is that the appropriate way to treat that variable? Likewise, 25(OH)D was a weighting factor in analyses because it was associated with conceiving. Is that appropriate to use the primary exposure variable also as a weighting factor in this sort of analyses?

Reviewer #3 (Remarks to the Author):

REVIEW OF PURDUE-SMITHE ET AL: The role of maternal preconception vitamin D status etc.

NOTES TO AUTHORS This is interesting, but, as I remark to the Editor, you present the data in (what to me is) an unnecessarily complex style. I would further recommend two points in connection with the Trivers-Willard hypothesis. 1. You cite Catalano but you do not mention that he and his colleagues report, not simply that maternal stress (presumably via maternal adrenal androgens) causes male foetuses to perish, but that frail male foetuses are selectively lost. This emphasises the evolutionary connection. 2. You do not cite the hypothesis of James that parental hormone levels around the time of conception are causally associated with offspring sex, high levels of testosterone being associated with male offspring. James has shown that this hypothesis supports the Trivers-Willard hypothesis (James 2013).

REFERENCES Catalano R, Bruckner T. 2006. Secondary sex ratios and male lifespan: damaged or culled cohorts. *Proc. Nat. Acad. Sci. USA* 103, 1639-43 James WH. 2013. Evolution and the variation of mammalian sex ratios at birth: reflections on Trivers and Willard (1973). *Journal of Theoretical Biology* 334, 141-148

Reviewer #4 (Remarks to the Author):

The authors examine whether the male to female sex birth ratio is lower when serum vitamin D levels are lower in data re-purposed from a recent randomized experiment.

1. Data on 1191 women is available. How much data was missing? 134, as stated deep in results?
2. Page 4, line 92. For all linear fits, e.g. 8% increase in male pregnancies, please show a scatterplot with the linear fit and a nonparametric (NP) smoother (at multiple smoothness levels). A carpet of the vitamin D levels would also be helpful. This can be in an appendix if the NP smoother supports the chosen linear fit.
3. Is this a standard cut point for insufficiency? If so, please say so and reference the source. If not, please argue why this cut point was chosen. If a set of cut points were examined both below and above this cut point, what are the results? A figure would be nice but could be relegated to an appendix if it supports the use of this cut point (or this cut point was determined a priori, such a claim would be needed to be stated explicitly by the authors). Imagine you are being deposed. How did you choose this cut point?
4. Page 9, line 221. Selection bias may occur similarly due to missing (not just lost) data.
5. Could one use the coefficients of variation for vitamin D levels to correct for random measurement error? Do such coefficients being >0 suggest there might be some dampening of the associations you present between vitamin D and sex ratio?
6. Was sex ratio obtained in a fashion masked to the vitamin D status?
7. Could there be reverse causation here? If vitamin D levels are obtained after the sex of the pregnancy is set. I remain somewhat unclear on the timing of vitamin D assessments relative to fetal development.
8. Page 13, line 336. Are continuous variables included using flexible modeling strategies like splines? Are first-order product terms included. Please provide some details about the IP weights, like mean, min, max, 1 and 99 percentiles.
9. Page 15. PLEASE do not state or use a cut point for statistical significance. Report findings (point estimates or bounds) that you believe are important, along with measures of uncertainty, like compatible (aka confidence) intervals or posterior probability distributions. (Perhaps report all P values to 3 decimal places?)

Reviewers' comments:

Reviewer #1 (Remarks to the Author):

“The role of maternal preconception vitamin D status in human offspring sex ratio”

This manuscript examines vitamin D status measured a varying amount of time prior to conception with offspring sex ratio measured both in pregnancies that are eventually lost and pregnancies that are delivered as live births. The results suggest that male conception is more likely in mothers who have sufficient vitamin D and that this association is stronger among women with higher levels of inflammation (measured by high sensitivity CRP). A strength of the paper is that the authors attempt to answer their research question with several analytical models, which supports the robustness of their results. Other comments meant to clarify the methods and results are presented below.

Response: We thank the reviewer for their positive remarks and thoughtful review of our paper. We hope that we have adequately addressed each of their questions and concerns in our point-by-point response below as well as our overall revisions to the manuscript.

1. It appears that this analysis has only included linear and dichotomous parameterizations of 25(OH)D. It would be interesting to see a more flexible modeling strategy to determine the shape of the association, i.e. with splines, or multiple categories, or a quadratic function, etc.

Response: We appreciate the reviewer’s suggestion to evaluate possible non-linear associations of vitamin D with male live birth. In response, we examined the possibly non-linear relationship between 25(OH)D concentrations and male live birth non-parametrically utilizing restricted cubic spline models and importantly, found no evidence of non-linearity. Specifically, in separate models, we specified 3, 4, and 5 knots, and evaluated the individual spline term contributions to the model fit and overall test for nonlinearity. The results from these models indicate no departures of linearity (*P*-values for overall tests of non-linearity all >0.48 for models with 3, 4, and 5 knots, respectively). We added the following details regarding this analysis on lines 81-82 and 344-347 in the text: “Results from restricted cubic spline models supported the linear fit of models evaluating continuous 25(OH)D and male live birth (Figure 2)” and “We examined the possibly nonlinear relationship between continuous 25(OH)D and male live birth non-parametrically using restricted cubic spline models (with 3 to 5 knots specified) and evaluated the individual spline term contributions to the model fit and overall test for nonlinearity.” In further response to this point and similar questions raised by Reviewers #2 and #4, we have also added a figure (Figure 2) below and to the manuscript, which illustrates the linearity of the 25(OH)D-male live birth relationship.

Figure 2. (a) Rug plot (dark blue) of preconception 25(OH)D levels with kernel density estimates for multiple bandwidth smoothers ($h = 1, 2, \text{ and } 3$); (b) Scatterplot of vitamin D levels (dark blue) and probability of male live birth with regression line and 95% confidence cloud

2. I wonder if the paper should exclude twins? Twins are detectable early in gestation and presumably the controls and influences on sex ratio in monozygotic or dizygotic twins could be different than the influences on a singleton pregnancy. Do the authors have evidence that vitamin D would have the same influence on the sex ratio of twin gestations as singleton gestations?

Response: The reviewer raises an excellent point. To address their concern, we conducted a sensitivity analysis in which we excluded twin gestations ($n=8$) from analyses that evaluated male live birth as the outcome. The results from this analysis were essentially identical (adjusted RR = 1.25, 95% CI = 1.03, 1.52) to the main analysis (RR = 1.25, 95% CI = 1.03, 1.52), suggesting that the inclusion of twins in our analyses does not change the overall results or interpretation of our data. We have included the following details regarding this sensitivity analyses on lines 121-123 and 391-393 in the text: “Estimates comparing male live birth in vitamin D sufficient versus insufficient women were similar in analyses restricted to singleton pregnancies (adjusted RR = 1.25, 95% CI = 1.03, 1.52).”

3. Vitamin D is measured at enrollment, prior to randomization, and then women try to conceive for a certain amount of time and then if they conceive their pregnancies are examined for sex of the offspring. Given that 25(OH)D may change over time, and that one proposed mechanism of action is uterine preparation and implantation, I wondered if there is misclassification of the exposure, possibly differential? Perhaps I’ve missed it, but did the authors consider some form of imputation that would estimate the 25(OH)D level at conception rather than using the baseline measure alone?

Response: It is true 25(OH)D may change over time, potentially contributing to misclassification of exposure. However, given that 25(OH)D was measured at the beginning of the cycle of the first pregnancy attempt after enrollment, and the primary outcome of interest is subsequent male live birth, this misclassification is unlikely to be differential. In other words, error in 25(OH)D

(the exposure), is unlikely to be related to the outcome (male live birth) because the offspring hadn't yet been conceived at the time of blood draw, and the laboratory that measured 25(OH)D in the preconception samples was blinded to the outcome of the pregnancy. Regarding the magnitude of variability in 25(OH)D levels over time, we would like to further clarify that most women (75%) became pregnant in the first 3 cycles following the baseline serum 25(OH)D assessment. The percent of pregnancies occurring in cycles 1-3 were 33%, 25%, and 17%, respectively. Thus, substantial variability in 25(OH)D levels from baseline to the conception cycle is unlikely. In sum, exposure misclassification in our study is likely: 1) non-differential and 2) not substantial. We have added the following details to the text:

Lines 222-225: “Finally, measurement error in 25(OH)D concentrations may have resulted in some degree of misclassification of vitamin D status; however, given that blood samples were collected prior to conception and laboratory measurement of 25(OH)D was blind to the outcome, we expect this misclassification to be non-differential.”

4. Why are both risk ratios and risk differences presented? The results section doesn't seem to distinguish their interpretation and I don't think having both provides additional information, while it does make the tables more crowded. They could be moved to a supplement if the authors wish to keep them, and if they prefer them in the main paper, they should add some rationale or interpretation for each set of estimates.

Response: We opted to include the risk differences to provide an estimate of the absolute number of additional male live births attributable to vitamin D sufficiency, relative to insufficiency. The interpretation, for example, “vitamin D sufficiency versus insufficiency was associated with a difference of 5.55 male live births per 100 women” indicates a clinically meaningful difference between the two groups. We believe that the risk differences provide a basis for the readership for interpreting practical significance; therefore, we request to provide both estimates, but have relegated the RDs for the primary analyses to **Supplementary Table 1**. We also clarified the interpretation of the risk difference on lines 74-75: “In absolute terms, vitamin D sufficiency versus insufficiency was associated with a difference of 5.55 male live births per 100 women (adjusted RD per 100 women = 5.55; 95% CI = 0.34, 10.67)”

5. Each study site probably varies in altitude, diet, air pollution, culture, climate, etc. Was study site related to sex ratio? And could it be a confounder of this analysis (or a surrogate for these underlying unmeasured factors)?

Response: The reviewer's point about potential confounding by study site/geographic location is an excellent one. Indeed, though there are small site differences in live birth, these do not seem to explain the findings with vitamin D. When additionally adjusting for site in our regression models we see similar results. Specific details regarding these analyses are shown below.

1) The proportions male live births for each study site were as follows: Buffalo, 22%; Scranton, 27%; Salt Lake City, 27%; Denver, 25%.

Interestingly, the data above suggest that the proportion of male live birth was lowest at the Buffalo site, which has northernmost line of latitude (42°), compared to other sites (Scranton, 41°; Salt Lake City, 41°; Denver, 40°). These data are in line with ecologic data suggesting reduced sex ratios at higher lines of latitude, as cited on lines 40-44 in the introduction.

2) We then conducted an additional analysis in which we further adjusted for study site and other potential confounding factors in our multivariable model. The vitamin D (sufficient versus insufficient) estimates from this model were virtually unchanged (adjusted RR = 1.23, 95% CI = 1.01, 1.51). In light of these data, confounding by site location does not appear to be a possible explanation for our positive findings.

To fully address this question and our additional analyses in the manuscript, we have added the following details to the text on lines 123-125: “Estimates for male live birth comparing vitamin D sufficient versus insufficient women were also robust to further adjustment for season of blood draw, study site, multivitamin use, and waist-to-hip ratio both in overall (adjusted RR = 1.23, 95% CI = 1.01, 1.51) and in stratified analyses (hsCRP >1.95 ng/mL adjusted RR = 1.51, 95% CI = 1.06, 2.14; hsCRP ≤1.95 ng/mL RR = 1.10, 95% CI = 0.87, 1.39).”

6. Relatedly, neither season nor vitamin use was associated with 25(OH)D in this population. Do the authors have any comment on this?

Response: The lack of association of 25(OH)D with vitamin use may be attributable to 93% of women using vitamins at enrollment. We agree that it is somewhat surprising that season of blood draw was not associated with 25(OH)D in this population. However, we expect seasonal variability of 25(OH)D levels in our study population to be somewhat lower than in the general population due to their being healthy, vitamin D replete, and taking multivitamins. To address the reviewer’s concern about potential confounding by season, we further adjusted for season in our multivariable model, which yielded estimates that were nearly identical to the main analysis (adjusted RR = 1.23, 95% CI = 1.01, 1.51). We have updated the text on lines 123-125 to read as follows: “Estimates for male live birth comparing vitamin D sufficient versus insufficient women were also robust to further adjustment for season of blood draw, study site, multivitamin use, and waist-to-hip ratio both in overall (adjusted RR = 1.23, 95% CI = 1.01, 1.51) and in stratified analyses (hsCRP >1.95 ng/mL adjusted RR = 1.51, 95% CI = 1.06, 2.14; hsCRP ≤1.95 ng/mL RR = 1.10, 95% CI = 0.87, 1.39).”

7. Induced abortion was likely rare in this population since they were trying to become pregnant, but can the authors confirm this in the methods?

Response: Thank you - we have added this detail to the methods on line 285: “There were no elective terminations during follow-up.”

8. What is the rationale for the variable “Number of previous pregnancies, not including losses”? This does not correspond to clinical gravidity and it is not clear, given the women in this study had a varying number of previous losses, why the losses are not considered conceptions?

Response: We agree with the reviewer and have deleted this variable from Table 1 and replaced it with a variable for parity.

9. Please describe if there were any values lower than the limit of detection for hsCRP. Also, add to the results the range of hsCRP values detected in the analysis sample.

Response: The range of detected hsCRP values was 0.15-63.67 ng/mL. We have added information on the distribution of hsCRP to the text on line 63, "Median preconception hsCRP was 1.22 ng/mL (IQR = 0.57-3.26 ng/mL)" and the limit of detection on line 305, "There were 21 values below the limit of detection of 0.15 ng/mL for hsCRP." Also of note, hsCRP data were only used to create categories of low-grade chronic inflammation vs. not-inflamed, using a cut-point derived previously with exclusion of very high values indicative of acute infection (also addressed in response to Reviewer #2's minor comment #2).

10. "Risk calculation using this denominator reflects the true natural process in a given population of women attempting to conceive because all women are "at risk" of having a live-born male at the beginning of follow-up, though some will experience a pregnancy loss or may not achieve a pregnancy at all." How can a non-pregnant woman be "at risk" of having a male birth? If she is infertile, she can't be at risk of any birth. I think I understand the concept that women who appear to have not conceived may have actually conceived and had an undetectable loss, and perhaps that happens more often for male conceptuses, but I don't quite understand how this can be estimated in these data since none of that is observed?

Response: In this setting, we are assuming that all women at 'at-risk' because they are all actively attempting to conceive and have no history of infertility. In conceptualizing the risk calculations using the different denominators it is helpful to think of this in the context of a set of hypothetical randomized trials (as each answers a slightly different question). The risk calculation for male live birth among all women who enrolled in the study is analogous to an intent-to-treat analysis that might be conducted if this were a randomized trial of preconception vitamin D status on male live birth. In this hypothetical trial, non-pregnant women who have demonstrated fecundability and are attempting to conceive would be enrolled, randomized to a vitamin D sufficient or insufficient arm, and then followed for a prespecified duration of time for male live birth. In the intent-to-treat analysis, we would estimate the RR of male live birth comparing the vitamin D sufficient versus insufficient arms among all women who were randomized. The only difference between this hypothetical trial and our observational study is, of course, the absence of randomization of vitamin D status at baseline to achieve exchangeability between vitamin D sufficient and insufficient groups (which we addressed by covariate adjustment in multivariable models). Additionally, 11% of women did not complete follow-up, and so we used inverse-probability weights to address potential selection bias. The interpretation of our inverse-probability weighted analyses among all women who completed follow up is thus, the association of preconception vitamin D status and male live birth that would have been observed if all women enrolled in EAGeR had completed follow-up. The estimates from analyses among pregnancies are interpreted as the association of preconception vitamin D status and male live birth that would have been observed if all women had completed follow-up and had an hCG-detected pregnancy. This analysis answers the question, "What is the effect of preconception vitamin D status on male live birth, given successful implantation?" Finally, the weighted analyses among live births describe the association of preconception vitamin D status and male live birth that would have been observed if all women had completed follow-up and had an hCG-detected pregnancy that survived to birth. This analysis answers the question, "What is the effect of the preconception vitamin D status on having a male baby, given that the infant was born alive?"

We hope that we have clarified our approach in the following text:

Lines 181-186: “The consistency of the estimates for vitamin D status and male live birth in our weighted analyses among all women who completed follow-up, pregnancies, and live births, in conjunction with the estimates for vitamin D and pregnancy with a male conceptus, suggest that the influence of preconception vitamin D status on the eventual live birth of a male begins at least as early as implantation, but does not necessarily rule out the potential for vitamin D to also act on male embryos after implantation via pregnancy loss.”

Lines 324-330: “Risk calculation using this denominator reflects the true natural process in a given population of women with demonstrated fecundability who are attempting to conceive, because at the beginning of follow-up, all women are “at risk” of eventually having a live-born male, though some will experience a pregnancy loss or may not achieve a hCG-detectable pregnancy at all. This approach is analogous to the intent-to-treat analysis of a hypothetical trial in which women are randomly assigned to sufficient or insufficient vitamin D status at preconception and then followed for male live birth for the prespecified duration of the trial.”

Lines 349-352: “The estimates from these pregnancy-weighted analyses are interpreted as the association of preconception vitamin D status with male live birth that would have been observed if all women who enrolled in EAGeR had completed follow-up and had an hCG-detected pregnancy.”

Lines 358-360: “The live birth-weighted estimates from these analyses are interpreted as the association of preconception vitamin D status with male live birth that would have been observed if all women in the study had completed follow-up and had an hCG-detected pregnancy that survived to birth.”

11. Are the results of the primary analysis changed if non-White women are excluded from the analysis? I understand that race was an adjustment factor in the model, but this may not fully account for differences by race.

Response: Estimates from models excluding non-White women were essentially the same as those from the main analysis (adjusted RR=1.27, 95% CI = 1.04, 1.54). We have added these details to the text:

Lines 116-118: “Associations for vitamin D status and male live birth were essentially unchanged in analyses restricted to White women (adjusted RR = 1.27, 95% CI = 1.04, 1.54).…”

Lines 377-379: “Second, to evaluate whether associations varied according to multiple gestation and race/ethnicity, we conducted analyses restricted to singleton pregnancies and to White women.”

12. Figure 1 is difficult to follow in terms of whether it is by participant or by gestations. For example, there are N=732 confirmed pregnancies, but in the boxes below 603+135+8 does not equal the above total of 732. Even if I account for twins, if my math is correct, 597+133+8 does not add up to 732. So, I’m having trouble following the Ns through this figure. Perhaps check the math throughout and label the units on each N to clarify whether it is participants or gestations in each box.

Response: We have revised the figure to clarify the participant flow. The source of the confusion seems to be related to the categorization of 6 ectopic pregnancies that occurred during follow-up. To clarify, the 732 “confirmed pregnancies” in the original flowchart did not include these 6 ectopic pregnancies, which were visualized by ultrasound, but were obviously not viable (so while these were clinically confirmed losses, they were not clinically confirmed pregnancies as no gestational sac was observed). We have updated the flowchart to indicate that these 6 ectopic pregnancies were in addition to the 732 clinically-confirmed pregnancies (of which 8 were twin gestations) and provided an explanation in a footnote to Figure 1. After accounting for these 6 ectopic pregnancies, the math is as follows: 732 confirmed pregnancies + 6 ectopic pregnancies + 8 twins = 8 lost to follow-up, 603 live born infants (which includes 6 twins, or total of 597 singleton gestations births and 3 twin gestation births) + 135 clinical losses (which includes 2 twins).

13. Introduction (lines 59-62): “Indeed, a large randomized placebo-controlled trial recently demonstrated that preconception-initiated low-dose aspirin administration restored a reduced sex ratio at birth among women with increased inflammation, providing direct evidence of this phenomenon in humans.” Is this the same study that is analyzed in this paper? If so, it should be clarified that the human evidence for inflammation influencing the sex ratio is limited to this one cohort.

Response: We have updated the text on line 35 as the reviewer suggests: “Indeed, we demonstrated previously in the Effects of Aspirin in Gestation and Reproduction [EAGeR] trial that preconception-initiated low-dose aspirin (LDA) administration restored a reduced sex ratio at birth among women with increased inflammation, providing direct evidence of this phenomenon in humans.”

14. “Vitamin D appears to regulate placental inflammation and may be involved in the maternal-fetal immunologic response necessary for fetal survival.¹⁹” Is reference 19 the correct reference? It seems to say that vitamin D may be important for cellular differentiation (particularly in the uterus) with evidence from human cell lines. I don’t immediately see a connection between that and “placental inflammation”? Could you explain further?

Response: Thank you for pointing out this missing reference. We have updated our reference list and provided the correct in-text citation () for this statement and clarified our language in the text on lines 47-48: “Vitamin D appears to regulate placental inflammation and may be involved in the immunologic response necessary for successful embryo implantation and survival.^{19, 20}”

19. Liu, N. Q. *et al.* Vitamin D and the Regulation of Placental Inflammation. *The Journal of Immunology* **186**, 5968-5974, doi:10.4049/jimmunol.1003332 (2011).

20. Du, H., Daftary, G. S., Lalwani, S. I. & Taylor, H. S. Direct regulation of HOXA10 by 1, 25-(OH) 2D3 in human myelomonocytic cells and human endometrial stromal cells. *Molecular Endocrinology* **19**, 2222-2233 (2005).

15. Does the lab that measured the 25(OH)D participate in DEQAS or the vitamin D standardization program?

Response: We appreciate the reviewer's question about vitamin D assay standardization/external reliability assessment. While the lab (lab of Dr. Michael Tsai, University of Minnesota) that measured our samples participates in DEQAS for other vitamin D assays (e.g., 1-25-dihydroxyvitamin D), the specific 25(OH)D immunoassay used in our analysis did not undergo DEQAS assessment. We assume that the reviewer's concern relates to potential measurement error in 25(OH)D concentrations, which is certainly possible. However, this measurement error is unlikely to be related to the outcome (male live birth/pregnancy with a male fetus), for reasons specified in response to their point #2 above.

16. What is the correlation between 25(OH)D and hsCRP?

Response: We added this detail to the text on lines 84-85: "The Spearman rank correlation coefficient between baseline 25(OH)D and hsCRP was -0.17 (95% CI = -0.23, -0.12)."

17. I would find it helpful if the Discussion would explain why male conception is preferred when the mother is "healthy"? And how would inflammation selectively target male conceptuses?

Response: Our interpretation of the literature is that a typical/expected male to female offspring sex ratio slightly favors males and then is reduced under unfavorable conditions. In other words, males may be reduced (as we observed previously with inflammation), and then, as observed in this paper, that ratio is restored by vitamin D sufficiency. To address this concept more clearly in the paper, we have added details and further explanation:

Lines 29-30: "According to evolutionary theory, the ability of some animal species to adjust their offspring sex ratio in response to maternal health status and resource availability confers a natural selection advantage, and under unfavorable conditions, females will be less likely to bear sons."

Lines 140-162: "To our knowledge, this study is the first to evaluate preconception vitamin D status and offspring sex ratio, though the notion that maternal nutrition and other factors may influence offspring sex ratio has notable precedent in the evolutionary biology literature. According to the Trivers-Willard sex allocation hypothesis, natural selection should favor species capable of shifting their offspring sex ratio in response to environmental stressors and maternal body condition in order to maximize the likelihood of passing on genetic material to grandchildren.¹ Across mammalian species, males exhibit greater variance in terms of lifetime reproductive success, with much of this variance explained by specific advantageous physical and social characteristics.²² For male offspring to eventually join the select group exhibiting these characteristics, greater parental investment is theoretically required relative to female offspring; therefore, when conditions are poor, male offspring will be less likely than female offspring to eventually successfully reproduce. Conceptually, an underlying distribution of survivability of male embryos exists in any given population, and the threshold of maternal tolerance to those at the lower tail of this distribution apparently shifts downward with exposure to population stressors, selectively increasing the rate of loss among male embryos with lower survivability likelihood.²³ In line with this theory, dramatically reduced secondary sex ratios in human populations have been reported during disasters¹¹, wars²⁴, and times of economic depression²⁵, and recent randomized clinical trial data implicate inflammation as a possible biologic pathway.⁹ Less is known about the influence of maternal nutrition on the human sex ratio, but experimental data from ruminants, rodents, marsupials, primates indicate that administration of low-calorie and low-fat diets, caffeine, and other dietary factors reduce the likelihood of male progeny.²² Our

results expand this body of research, suggesting that maternal nutrition - specifically, vitamin D status - may play a role in the offspring sex ratio of humans, possibly by “rescuing” male embryos that may have otherwise been lost in women with elevated inflammation.”

Lines 189-200: “In terms of sex-specific probability of implantation, bovine and ovine preimplantation embryos exhibit sexually dimorphic secretion of interferon-tau, the cytokine necessary for uterine receptivity signaling to sustain implantation.³⁰ with female bovine preimplantation embryos producing nearly twice as much interferon-tau as their male counterparts.³¹ In humans, hCG secretion by preimplantation embryos similarly contributes to maternal immunotolerance required for successful implantation.³² Although data on sex differences of hCG production by human preimplantation embryos are unavailable, some evidence suggests small differences in hCG concentrations in pregnancies with surviving male versus female embryos during the week following implantation.³³ If lower hCG secretion by male preimplantation embryos confers higher susceptibility to preimplantation failure among women with elevated inflammation, adequate 25(OH)D levels may provide some remediation, thereby increasing the likelihood of survival and eventual implantation of male embryos. Our observed stronger positive associations for vitamin D status and pregnancy with a male fetus among women with elevated inflammation lend support to this possibility, though other physiologic mechanisms, such as those involving sex hormone levels are possible.³⁴”

Minor comments

Page 11, line 282, I believe “Ashville” is spelled incorrectly.

Response: Thank you; we have corrected this typographical error (line 295).

Tables 2 and 4 have EAGeR dates as 2007-2012, Table 1 has 2007-2011, and the text (line 260) had 2007-2011.

Response: Thank you - this been updated to “2007-2011” throughout.

Table 2 has column header “Model 1,” but this could be replaced with the header “Unadjusted” which is more informative. Similarly, “Model 2” and “Model 3” could be replaced with “Adjusted” with their corresponding footnotes to distinguish the adjustment sets.

Response: We have renamed our models (i.e., unadjusted and adjusted) in the tables and throughout the text, as the reviewer suggests. To simplify our tables, we removed adjusted model 2 from Table 2 (given that estimates were essentially identical to model 3) and specified in the text on lines 118-122 that further adjustment for other factors made no meaningful difference in our estimates: “Estimates for male live birth comparing vitamin D sufficient versus insufficient women were also robust to further adjustment for season of blood draw, study site, multivitamin use, and waist-to-hip ratio both in overall (adjusted RR = 1.23, 95% CI = 1.01, 1.51) and in stratified analyses (hsCRP >1.95 ng/mL adjusted RR = 1.51, 95% CI = 1.06, 2.14; hsCRP ≤1.95 ng/mL RR = 1.10, 95% CI = 0.87, 1.39).”

Did any women enroll in EAGeR twice? Did any women have two pregnancy attempts in the study?

Response: To clarify, women were only eligible to participate in the study once. However, a small number (N = 45) of women conceived more than once during the 6 menstrual cycles of initial follow-up (e.g. experienced a loss followed by another pregnancy attempt). Analyses are limited to the final outcome observed for these women.

In Table 2 the N's aren't totally clear. The figure shows 292 male births total, but in Table 2 adding all of the male births in the hsCRP analysis, for example, results in 293 male births (88+111+48+46). Is that a typo?

Response: Thank you for your attention to this detail. This was a typographical error – the 46 was supposed to be 45. Table 2 is now corrected.

Page 8, line 185, “LDA” is used here, although it was not defined as an acronym when the phrase was first used in the introduction.

Response: Thank you for pointing this out. We have indicated that “LDA” refers to “low-dose aspirin” at its first use on line 36.

Reviewer #2 (Remarks to the Author):

Review NCOMMS-19-11592: The role of maternal preconception vitamin D status in human offspring sex ratio.

The authors conducted a secondary analysis of data from the EAGeR study, a trial of pre-conception initiated low-dose aspirin treatment to investigate effects on live birth rate. The initial eligibility criteria required that participants had a single prior pregnancy loss (occurring at less than 20 weeks gestation and within the last year). A later, expanded eligibility criteria allowed for enrollment of women with one or two prior losses (loss could occur at any gestational age and anytime in past). This submitted manuscript addressed the question of whether preconception 25(OH)D concentrations were associated with sex ratio of the offspring (live births). The authors report that women with preconception 25(OH)D ≥ 30 ng/ml were more likely than the comparison group (women with lower 25(OH)D) to have boy babies. The association was stronger in women with C-reactive protein >1.95 ng/ml (a marker of chronic inflammation).

This is a novel finding, and sex-ratio data tend to be of general interest. Unfortunately, sex-ratio findings rarely replicate.

The analytic approach was to start with the preconception cohort and use inverse probability weighting (IPW) to account for loss to follow-up, for the large subgroup that did not conceive an identifiable pregnancy, and for pregnancy losses before live birth. This is not the traditional approach to analyzing sex-ratio data. The author's approach relies on being able to predict those transitions adequately with available data to assign appropriate weights. It is not clear how well that can be done. More importantly, the benefits of doing this are unclear. While it provides results that may be more representative of the

study participant sample, the participants are volunteers with prior pregnancy losses, recruited under two different eligibility criteria. Thus, the external population to whom this is generalizable is uncertain. Also, the transition from trying to become pregnant to being pregnant and having a live birth is a fundamental state change. The sex of the baby is not a property of the woman until she is pregnant and can identify the sex. So, it seems inappropriate to start with the full study sample.

Response: We thank the reviewer for their careful review of our paper and hope that we have satisfactorily answered each of their questions in our response. The reviewer's overall questions regarding our statistical and methodological approach, external generalizability of our findings, issues of non-replicability of findings in the sex ratio literature are well-taken. We would like to respond to each of these three main points and hopefully provide more clarity.

1) Statistical and methodologic approach:

In this setting, we are assuming that all women at 'at-risk' because they are all actively attempting to conceive and have no history of infertility. In conceptualizing the risk calculations using the different denominators it is helpful to think of this in the context of a set of hypothetical randomized trials (as each answers a slightly different question). The risk calculation for male live birth among all women who enrolled in the study is analogous to an intent-to-treat analysis that might be conducted if this were a randomized trial of preconception vitamin D status on male live birth. In this hypothetical trial, non-pregnant women who have demonstrated fecundability and are attempting to conceive would be enrolled, randomized to a vitamin D sufficient or insufficient arm, and then followed for a prespecified duration of time for male live birth. In the intent-to-treat analysis, we would estimate the RR of male live birth comparing the vitamin D sufficient versus insufficient arms among all women who were randomized. The only difference between this hypothetical trial and our observational study is, of course, the absence of randomization of vitamin D status at baseline to achieve exchangeability between vitamin D sufficient and insufficient groups (which we addressed by covariate adjustment in multivariable models). Additionally, 11% of women did not complete follow-up, and so we used inverse-probability weights to address potential selection bias. The interpretation of our inverse-probability weighted analyses among all women who completed follow up is thus, the association of preconception vitamin D status and male live birth that would have been observed if all women enrolled in EAGeR had completed follow-up. The estimates from analyses among pregnancies are interpreted as the association of preconception vitamin D status and male live birth that would have been observed if all women had completed follow-up and had an hCG-detected pregnancy. This analysis answers the question, "What is the effect of preconception vitamin D status on male live birth, given successful implantation?" Finally, the weighted analyses among live births describe the association of preconception vitamin D status and male live birth that would have been observed if all women had completed follow-up and had an hCG-detected pregnancy that survived to birth. This analysis answers the question, "What is the effect of the preconception vitamin D status on having a male baby, given that the infant was born alive?"

We have clarified the meaning of each of these analyses in the text:

Lines 181-186: "The consistency of the estimates for vitamin D status and male live birth in our weighted analyses among all women who completed follow-up, pregnancies, and live births, in conjunction with the estimates for vitamin D and pregnancy with a male conceptus,

suggest that the influence of preconception vitamin D status on the eventual live birth of a male begins at least as early as implantation, but does not necessarily rule out the potential for vitamin D to also act on male embryos after implantation via pregnancy loss.”

Lines 324-330: “Risk calculation using this denominator reflects the true natural process in a given population of women with demonstrated fecundability who are attempting to conceive, because at the beginning of follow-up, all women are “at risk” of eventually having a live-born male, though some will experience a pregnancy loss or may not achieve a hCG-detectable pregnancy at all. This approach is analogous to the intent-to-treat analysis of a hypothetical trial in which women are randomly assigned to sufficient or insufficient vitamin D status at preconception and then followed for male live birth for the prespecified duration of the trial.”

Lines 349-352: “The estimates from these pregnancy-weighted analyses are interpreted as the association of preconception vitamin D status with male live birth that would have been observed if all women who enrolled in EAGeR had completed follow-up and had an hCG-detected pregnancy.”

Lines 358-360: “The live birth-weighted estimates from these analyses are interpreted as the association of preconception vitamin D status with male live birth that would have been observed if all women in the study had completed follow-up and had an hCG-detected pregnancy that survived to birth.”

- 2) External generalizability of findings: Pregnancy loss is an extremely common event (31% of all hCG-detected pregnancies do not survive) (Wilcox et al) and although Recurrent Pregnancy Loss (>2 losses) (Ford et al) is considered a pathological condition, having 1-2 pregnancy losses is not. In response to the reviewer’s point regarding eligibility criteria strata (1 vs. 2 prior losses) in their point #3 below, eligibility criteria-stratified analyses revealed stronger associations among women with 1 prior loss than among women with 2 prior losses. We believe that these results lend support to the generalizability of our findings, in that this subgroup of women are the most representative of the general population in terms of their history of loss.

Wilcox AJ et al. Incidence of early loss of pregnancy. *N Engl J Med.* 1988 Jul 28;319(4):189-94.

Ford et al . Recurrent pregnancy loss: etiology, diagnosis, and therapy. *Rev Obstet Gynecol.* 2009 2(2):76-83.

- 3) General lack of replicability in sex ratio literature: It may be true that there is a lack of consistent findings across studies on various factors influencing the human sex ratio. Inconsistencies may be due to a variety of factors, including differences in study design (ecologic versus prospective versus cross-sectional) and/or differences in the underlying prevalence and distribution of effect modifiers (e.g., inflammation) in study populations, though reconciling differences in findings for other exposures is outside the scope of our paper. Our study is based on high quality prospective data in a large clinical cohort. As it is

the first to evaluate vitamin D status and offspring sex ratio specifically, we cannot directly compare our findings to previous research.

Additional major concerns.

1. 25(OH)D was measured from samples collected prior to randomization, but women took from 1 to 6 cycles to conceive. Thus, for some, the conception occurred in a different season than blood collection, and 25(OH)D is seasonal, especially in higher latitudes. The likelihood of misclassification of 25(OH)D due to a single measurement will increase with time to pregnancy. All of the women who did not conceive would likely have different concentrations by the end of their study attempts than the measured values.

Response: We acknowledge that 25(OH)D may change over time, potentially contributing to misclassification of exposure. However, given that 25(OH)D was measured at the beginning of the cycle of the first pregnancy attempt after enrollment, and the primary outcome of interest is subsequent male live birth, this misclassification is unlikely to be differential. In other words, error in 25(OH)D (the exposure), is unlikely to be related to the outcome (male live birth) because the offspring hadn't yet been conceived at the time of blood draw, and the laboratory that measured 25(OH)D in the preconception samples was blinded to the outcome of the pregnancy. Regarding the magnitude of variability in 25(OH)D levels over time, we would like to further clarify that most women (75%) became pregnant in the first 3 cycles following the baseline serum 25(OH)D assessment. The percent of pregnancies occurring in cycles 1-3 were 33%, 25%, and 17%, respectively. Thus, substantial variability in 25(OH)D levels from baseline to the conception cycle is unlikely. In sum, exposure misclassification in our study is likely: 1) non-differential and 2) not substantial. We have added the following details to the text:

Lines 222-225: "Finally, measurement error in 25(OH)D concentrations may have resulted in some degree of misclassification of vitamin D status; however, given that blood samples were collected prior to conception and laboratory measurement of 25(OH)D was blind to the outcome, we expect this misclassification to be non-differential."

2. The data on sex of pregnancy losses is highly selected and unlikely to be representative of all losses. Only a minority of the losses could be sexed, and whether or not they can be sexed is highly dependent upon gestational age at loss, sample collection, and probably the time between embryonic/fetal death and sample collection. The authors talk about sex ratio at implantation, but this is unobservable. Therefore, it is difficult to agree that their data "suggest that vitamin C sufficiency versus insufficiency is positively associated with survival and implantation of a male embryo particularly among women with higher inflammation."

Response: The reviewer's point is well-taken and as noted on lines 397-399, we fully agree that the missingness of the fetal sex variable of the pregnancy losses is not random and is likely dependent on gestational age, sample collection, timing between the loss and its collection, and possibly other factors. In response, we also removed the sentence from the first paragraph of the discussion which emphasized the findings of analyses using the karyotype data. However, this specific conclusion was based on the totality of evidence (not just the karyotype analysis alone) and we have ultimately opted to retain this analysis in the paper because the karyotype data (albeit with missingness) on the pregnancy losses is a unique feature of this study that provides important information on timing of susceptibility of male and female embryos to vitamin D

insufficiency. These data support the findings from our three-denominator approach (all women with complete follow-up, all pregnancies and all live births) by suggesting that the influence of vitamin D on the eventual live birth of a male occurs very early on. We have clarified that the basis of these conclusions on lines 181-186: “The consistency of the estimates for vitamin D status and male live birth in our weighted analyses among all women who completed follow-up, pregnancies, and live births, in conjunction with the estimates for vitamin D and pregnancy with a male conceptus, suggest that the influence of preconception vitamin D status on the eventual live birth of a male begins at least as early as implantation, but does not necessarily rule out the potential for vitamin D to also act on male embryos after implantation via pregnancy loss.”

3. In the initial EAGeR trial results paper, the two recruitment methods yielded somewhat different results. Was that taken into account in this secondary analysis?

Response: To address the reviewer’s question, we stratified our main analyses by the original (1 recent prior loss) and expanded (1-2 prior losses at any gestational age and any time in the past) eligibility criteria for the trial. In doing so, we found that the associations were more pronounced in the original stratum (adjusted RR = 1.49, 95% CI = 1.12-1.96) compared to the expanded stratum (adjusted RR = 1.05, 95% CI = 0.78-1.40). Reasons for differences in findings among women across strata are unclear. However, compared to women in the expanded stratum, women in the original stratum had slightly lower BMI (25.8 vs. 26.8 kg/m²), lower hsCRP (2.6 vs. 3.2 ng/mL), and slightly lower 25(OH)D levels (30.3 vs. 31.2 ng/mL). The differences in characteristics between these two groups, namely BMI and hsCRP, suggest that the influence of vitamin D on male live birth may be stronger among women with non-obesity-related inflammation. We also postulate that a larger proportion of women in the stratum of 2 prior losses would ultimately go on to be characterized as having recurrent pregnancy loss (>2 prior losses) and may therefore have some different underlying pathology. Importantly, we believe that the stronger findings among women in the original stratum lend support to the generalizability of our findings, in that these women are more representative of the general population of women attempting pregnancy.

4. The reported interaction between 25(OH)D and CRP has associated p-values of greater than 0.23, with the exception of one which was 0.14, so this interaction, though interesting, is not a strongly supported finding.

Response: In response to the reviewer’s concern, we have softened our language regarding these stratified analyses on lines 137: “we observed somewhat stronger associations among women with elevated hsCRP...”, 143: “...suggesting that sufficient preconception levels of vitamin D may ameliorate an inflammatory process...”, and 208-209: “Our observed stronger positive associations for vitamin D status and pregnancy with a male fetus among women with elevated inflammation lend support to this possibility, though other physiologic mechanisms, such as those involving sex hormone levels are possible.” Given that effect modification by hsCRP on the vitamin D-male live birth relationship was specified *a priori*, as we hypothesized that the role of vitamin D on sex ratio might work through immunomodulatory effects, we would like to keep the stratified findings and our interpretation of them in the manuscript. We also believe this is an important result to present as it may be hypothesis-generating for future work. We recognize that tests for interaction require considerably higher statistical power than the testing of main effects

and that the p-values did not meet standard criteria. Given that the estimates for vitamin D sufficiency versus insufficiency among women with elevated inflammation versus low inflammation are meaningfully different, we would request to retain these stratified analyses, but have removed the language regarding reliance on significance testing.

Specific comments/questions

1. Number of prior losses should be included in Table 1, as well as recruitment criteria.

Response: We have added the number of prior losses and eligibility strata to Table 1, as the reviewer suggests.

2. The cut-point of 1.95 for CRP was at the highest tertile value. I am under the impression that 3.0 is often used in CVD risk. Is 1.95 a conceptually good cut-point? Also, were there women with such high levels of CRP that it is likely due to an acute health problem, not chronic inflammation? If so, how were these values dealt with?

Response: It is true that >3.0 ng/mL is often used to classify individuals at risk of CVD; however, to date, hsCRP cut-points relevant for pregnancy outcomes have yet to be established. To clarify, our cut-point used in the paper, 1.95 ng/mL, corresponds to the 3rd tertile of hsCRP in this population (Sjaarda et al) exclusive of women with values possibly indicative of acute illness/injury (>10 ng/L) (Nehring et al). We have added these details to the text on lines 372-374: “This cut-point corresponds to the highest tertile of hsCRP in the study population, after excluding women with values >10 mg/L, which may be indicative of acute inflammation resulting from injury or illness.⁵⁰”

In further response to the reviewer, we would like to note that a 2.0 ng/mL cut-point has also been used in the cardiovascular literature. The JUPITER trial used hsCRP >2.0 ng/mL to identify women with low-grade inflammation for statin intervention and importantly, found that statin use lowered risk of cardiovascular events among these women. (Ridker et al) To further reassure the reviewer that our findings are not particularly sensitive to hsCRP cut-point, we would like to note the following: among women with hsCRP >3.0 (n=319), the RR for male live birth comparing vitamin D sufficiency versus insufficiency was 1.45 (95% CI = 0.94, 2.23). Furthermore, the RRs for vitamin D sufficiency versus insufficiency and male live birth in hsCRP >1.95 group were 1.37 (95% CI = 0.93, 2.02), 1.45 (95% CI, 2.07), and 1.46 (95% CI = 1.02, 2.09), after excluding women with hsCRP values >10 , >15 , and >20 , respectively.

Sjaarda, L. A. et al. Preconception Low-Dose Aspirin Restores Diminished Pregnancy and Live Birth Rates in Women With Low-Grade Inflammation: A Secondary Analysis of a Randomized Trial. *J Clin Endocrinol Metab* **102**, 1495-1504, doi:10.1210/jc.2016-2917 (2017).

Nehring SM, Patel BC. C Reactive Protein (CRP) [Updated 2019 Apr 21]. In: StatPearls [Internet]. Treasure Island (FL): StatPearls Publishing; 2019 Jan-. Available from: <https://www.ncbi.nlm.nih.gov/books/NBK441843/>

Ridker PM et al., Rosuvastatin to Prevent Vascular Events in Men and Women with Elevated C-Reactive Protein. *N Engl J Med* 2008; 359:2195-2207.

3. Exactly how were twins entered into the analysis?

Response: The unit of analysis was the mother-offspring pair. 8 twin gestations contributed 16 observations, and generalized estimated equations accounted for correlated data within twin gestations. To further reassure the reviewer that the inclusion of twins would not have influenced the estimates, we conducted a sensitivity analysis excluding these 16 observations. The results from this analysis were essentially unchanged (vitamin D sufficient vs. insufficient adjusted RR = 1.25, 95% CI = 1.03, 1.52), suggesting that the inclusion of twins in our analyses does not change the overall results or interpretation of our data. We have included the following details regarding this sensitivity analysis on lines 116-118 in the text: “Associations for vitamin D status and male live birth were essentially unchanged in analyses restricted to White women (adjusted RR = 1.27, 95% CI = 1.04, 1.54) and to singleton pregnancies (adjusted RR = 1.25, 95% CI = 1.03, 1.52).””

4. Why was waste/hip ratio used instead of BMI as a covariate?

Response: We opted to include waist-to-hip ratio as a covariate because it better predicts whole body fat percentage and visceral adipose tissue mass than BMI. (Swainson et al) Further, prior research has indicated that WHR is more predictive of fecundability than BMI. (Wise et al) In response to the reviewer, we have added the Wise et al reference to the text on line 346.

References:

Swainson MG, Batterham AM, Tsakirides C, Rutherford ZH, Hind K (2017) Prediction of whole-body fat percentage and visceral adipose tissue mass from five anthropometric variables. *PLOS ONE* 12(5): e0177175.

Wise LA, Palmer JR, Rosenberg L. Body size and time-to-pregnancy in black women. *Hum Reprod.* 2013 Oct;28(10):2856-64. doi: 10.1093/humrep/det333.

5. The paragraph (lines 188-201) supporting male vs female differences in maternal recognition of pregnancy refer to non-human species. Mechanisms of maternal recognition of pregnancy vary dramatically within mammals. In humans, hCG is of primary importance. The only relevant data of which I am aware show no differences in the hCG rise during the week following implantation by sex at birth (Nepomnaschy et al., Human Reproduction, 2008).

Response: We appreciate the reviewer’s attention to this detail and for providing this reference. It is true that hCG secretion by the embryo, rather than interferon-tau, is the primary mechanism involved in uterine signaling in humans, and we have revised the paragraph on lines 187-200 accordingly. Regarding the provided reference, the Nepomnaschy et al study indicates a slight difference ($P = 0.09$) in mean hCG secretion by male ($n=65$) versus female ($n=56$) embryos in the first week following implantation, which we believe is notable given the small sample size. We would also like to point out that their analysis included only embryos that survived to clinical confirmation of the pregnancy (~6 weeks’ gestation) and would therefore not capture any potential sex differences in embryonic hCG secretion among those that did not survive. Given that our data suggest that vitamin D influences probability of male live birth beginning very early on in the reproductive process (sometime between fertilization and implantation), the Nepomnaschy et al data are not inconsistent with our findings and hypothesized biologic mechanism.

Please find the revised the text on lines 196-205: “In terms of sex-specific probability of implantation, bovine and ovine preimplantation embryos exhibit sexually dimorphic secretion of

interferon-tau, the cytokine necessary for uterine receptivity signaling to sustain implantation,³⁰ with female bovine preimplantation embryos producing nearly twice as much interferon-tau as their male counterparts.³¹ In humans, hCG secretion by preimplantation embryos similarly contributes to maternal immunotolerance required for successful implantation.³² Although data on sex differences of hCG production by human preimplantation embryos are unavailable, some evidence suggests small differences in hCG concentrations in pregnancies with surviving male versus female embryos during the week following implantation.³³ If lower hCG secretion by male preimplantation embryos confers higher susceptibility to preimplantation failure among women with elevated inflammation, adequate 25(OH)D levels may provide some remediation, thereby increasing the likelihood of survival and eventual implantation of male embryos. Our observed stronger positive associations for vitamin D status and pregnancy with a male fetus among women with elevated inflammation lend support to this possibility, though other physiologic mechanisms, such as those involving sex hormone levels are possible.”

6. Lines 218-219: Over the last 50 or more years there have been numerous publications reporting aberrant sex ratios associated with various factors, but consistent findings across several studies are generally lacking.

Response: Given that no prior studies, as we are aware, have evaluated vitamin D and offspring sex ratio specifically, we cannot compare our results to previous research. Prior studies of other factors may have yielded inconsistent findings, but we can only speculate reasons for why this may be. It is possible that inconsistencies may be due in part to differences in study design (ecologic versus prospective or cross-sectional) and/or differences in the underlying prevalence of effect modifiers (e.g., inflammation), though reconciling differences in findings for other exposures is outside the scope of our paper. Importantly, we would like to emphasize as strengths of our paper the large prospective design, biomarker exposure measurement, minimal loss to follow-up, and the robustness of our results to many sensitivity analyses.

7. Inter-assay CVs are given, but not intra-assay CVs. The 25(OH)D CV for the blinded control was 17% which is quite high.

Response: We added the intra-assay CVs, which are notably lower than the inter-assay CVs, to the revised manuscript: “For 25(OH)D, the intra-assay coefficients of variation were 8.2% at a concentration of 15.5ng/mL and 5.5% at a concentration of 41.6ng/mL for two manufacturer-lyophilized controls, and 5.6% at a concentration of 40.3 ng/mL for the pooled control.”

Regardless, measurement error in 25(OH)D (the exposure), is unlikely to be related to the outcome (male live birth) because the offspring hadn’t yet been conceived at the time of blood draw, and the laboratory that measured 25(OH)D in the preconception samples was blinded to the outcome of the pregnancy. Thus, the misclassification arising from measurement error in 25(OH)D levels is likely non-differential misclassification and is unlikely to explain our positive findings.

8. It would be helpful to list in a table the factors used in the IPW designed to account for each of the types of ‘missingness’ (able to conceive, conception survived to live birth, traditional loss to follow-up along the way, any others?). It was not easy to determine which factors were used for what.

Response: To clarify, the specific predictors used to generate each of the weights were: 25(OH)D (continuous), age (continuous), BMI (continuous), number of previous losses (1, 2, or >2), number of previous live births (0, 1, or >1), treatment arm (LDA vs. placebo), marital status (married or unmarried), season of blood draw (spring, summer, fall, winter). These same factors were used to generate each other weights (loss to follow-up, pregnancies, live births). Details can be found in the text:

Lines 343-344: “All models included stabilized inverse-probability weights to account for withdrawal or loss to follow-up during the study period, which were generated using the following factors: 25(OH)D, age, BMI, number of previous pregnancy losses, number of previous live births, LDA treatment group, marital status, and season of blood draw.”⁴⁷”

Lines 353-355: “Variables used to generate the weights included factors associated with pregnancy in this cohort (i.e., 25(OH)D, age, BMI, number of previous pregnancy losses, number of previous live births, LDA treatment group, marital status, and season of blood draw).”

Lines 358-359: “We also conducted these analyses among live births (n=601 including 6 twin gestations) and employed inverse-probability weights to account for the conditional probability of completed follow-up, pregnancy, and live birth. Predictors used to generate the weights were the same as those used to generate the pregnancy weights.”

9. The low-dose aspirin treatment of the trial (which can lower CRP) was treated as a weighting factor in IPW analyses. Was that done for all analyses? Is that the appropriate way to treat that variable? Likewise, 25(OH)D was a weighting factor in analyses because it was associated with conceiving. Is that appropriate to use the primary exposure variable also as a weighting factor in this sort of analyses?

Response: Yes – the inclusion of 25(OH)D and LDA in models used to generate inverse-probability weights is appropriate in the context our paper, as the goal of our use of these weights is to account for potential selection bias (not for time dependent confounding). To explain briefly, vitamin D status is associated with loss to follow-up, probability of pregnancy, and probability of live birth in our study population. Each of these groups of individuals are not exactly representative of the overall study population at baseline (n=1,228) in terms of vitamin D status, LDA treatment arm, and other factors. By conducting analyses that are restricted to each of these groups, we are inherently conditioning (selecting) on retention in the cohort (non-loss to follow-up), pregnancy, and live birth, in each of these analyses, respectively, which could induce selection bias if these selection factors are also specifically associated with male live birth. Essentially, the inverse-probability weights minimize this bias (under assumptions) by creating a “pseudo-population” in which the exposure (vitamin D status) and other predictors (e.g., LDA) are unrelated to the probability of selection into each of these groups (e.g., makes those who were lost to follow-up appear similar to those retained with regard to vitamin D status and other predictors of loss to follow-up). By using weights to minimize the variability in vitamin D levels that is associated with selection into each of these groups, we can then estimate the association of preconception vitamin D status on male live birth that would have been observed if all women had completed follow-up (weighted analyses among all women with complete follow-up), had completed follow-up and had an hCG-detected pregnancy (weighted analyses among hCG-detected pregnancies), and had completed follow-up and had a hCG-detected pregnancy that survived to birth (weighted analyses among live births).

Reviewer #3 (Remarks to the Author):

REVIEW OF PURDUE-SMITHE ET AL: The role of maternal preconception vitamin D status etc.

NOTES TO AUTHORS This is interesting, but, as I remark to the Editor, you present the data in (what to me is) an unnecessarily complex style.

Response: We, along with the editorial team, contend that the level of complexity in our analysis is appropriate for testing our hypotheses. However, in response to the reviewer, we simplified the presentation of our data by relegating the risk differences originally presented in Table 2 to Supplementary Table 1, and now present only the unadjusted and main multivariable model in Table 2, therefore allowing us to reduce the number of main tables from 5 to 4.

I would further recommend two points in connection with the Trivers-Willard hypothesis. 1. You cite Catalano but you do not mention that he and his colleagues report, not simply that maternal stress (presumably via maternal adrenal androgens) causes male foetuses to perish, but that frail male foetuses are selectively lost. This emphasises the evolutionary connection.

Response: Thank you for pointing this out. We have added the following text on lines 150-154 to include this information: "Conceptually, an underlying distribution of survivability of male embryos exists in any given population, and the threshold of maternal tolerance to those at the lower tail of this distribution apparently shifts downward with exposure to population stressors, selectively increasing the rate of loss among male embryos with lower survivability likelihood."²³

2. You do not cite the hypothesis of James that parental hormone levels around the time of conception are causally associated with offspring sex, high levels of testosterone being associated with male offspring. James has shown that this hypothesis supports the Trivers-Willard hypothesis (James 2013).

REFERENCES Catalano R, Bruckner T. 2006. Secondary sex ratios and male lifespan: damaged or culled cohorts. Proc. Nat. Acad. Sci. USA 103, 1639-43 James WH. 2013. Evolution and the variation of mammalian sex ratios at birth: reflections on Trivers and Willard (1973). Journal of Theoretical Biology 334, 141-148

Response: We have updated lines 199-200 to indicate that other mechanisms might also explain our findings and have included the James reference. The text now reads: "Our observed stronger positive associations for vitamin D status and pregnancy with a male fetus among women with elevated inflammation lend support to this possibility, though other physiological mechanisms, such as those involving sex hormone levels are possible."³⁴

Reviewer #4 (Remarks to the Author):

The authors examine whether the male to female sex birth ratio is lower when serum vitamin D levels are lower in data re-purposed from a recent randomized experiment.

1. Data on 1191 women is available. How much data was missing? 134, as stated deep in results?

Response: We have clarified the degree of missingness in the Methods and moved these details to the beginning of the Statistical Analysis section (lines 313-316): "Out of 1,228 women who

enrolled in the study, percentages of missing data ranged from 0.08% for physical activity to 3.58% for employment status for all predictor variables. Karyotype/fetal sex data was missing for N = 134 of 190 (71%) pregnancy losses and N = 2 of 603 (0.003%) live births.”

2. Page 4, line 92. For all linear fits, e.g. 8% increase in male pregnancies, please show a scatterplot with the linear fit and a nonparametric (NP) smoother (at multiple smoothness levels). A carpet of the vitamin D levels would also be helpful. This can be in an appendix if the NP smoother supports the chosen linear fit.

Response: We appreciate the reviewer’s suggestion and have provided the following figure (**Figure 2**), which shows: (a) the rug for vitamin D levels with kernel density estimates (with multiple bandwidth smoothers specified [$h = 1, 2,$ and 3]); (b) the scatterplot of vitamin D levels and overall probability of male live birth with the regression line and 95% confidence cloud depicting the linear fit.

Figure 2. (a) Rug plot (dark blue) of preconception 25(OH)D levels with kernel density estimates for multiple bandwidth smoothers ($h = 1, 2,$ and 3); (b) Scatterplot of vitamin D levels (dark blue) and probability of male live birth with regression line and 95% confidence cloud

3. Is this a standard cut point for insufficiency? If so, please say so and reference the source. If not, please argue why this cut point was chosen. If a set of cut points were examined both below and above this cut point, what are the results? A figure would be nice but could be relegated to an appendix if it supports the use of this cut point (or this cut point was determined a priori, such a claim would be needed to be stated explicitly by the authors). Imagine you are being deposed. How did you choose this cut point?

Response: We thank the reviewer for their attention to our vitamin D cut-point. We would like to clarify that our *a priori* dichotomization of ≥ 30 vs. < 30 ng/mL comes from the established Endocrine Society (Holick et al) definition of sufficiency vs. insufficiency. We have clarified our use of the Endocrine Society cut-point on lines 307-310 in the text: “Women were classified a priori as vitamin D insufficient (25(OH)D < 30 ng/mL) or sufficient (25(OH)D ≥ 30 ng/mL)

according to Endocrine Society cut-points for optimal bone health, as cut-points for reproductive health have yet to be established.”

Holick, M. F. *et al.* Evaluation, treatment, and prevention of vitamin D deficiency: an Endocrine Society clinical practice guideline. *J Clin Endocrinol Metab* **96**, 1911-1930 (2011).

4. Page 9, line 221. Selection bias may occur similarly due to missing (not just lost) data.

Response: We acknowledge that missing data may induce bias and have added text on line 220 to address the reviewer’s point: “Further, selection bias is possible if loss to follow-up or missingness is related to both vitamin D status and probability of pregnancy with a male or male live birth.”

5. Could one use the coefficients of variation for vitamin D levels to correct for random measurement error? Do such coefficients being >0 suggest there might be some dampening of the associations you present between vitamin D and sex ratio?

Response: We appreciate the reviewer’s suggestion to correct for measurement error in vitamin D levels; however, we are unable to locate any established methods that would accomplish this using inter-assay and intra-assay CVs. Importantly, although 25(OH)D levels are measured with error, we believe that any misclassification would be non-differential and would not explain our main findings, that preconception serum 25(OH)D is positively associated with male live birth. We have added language to acknowledge the potential impact of measurement error in 25(OH)D levels on lines 223-226 : “Finally, measurement error in 25(OH)D concentrations may have resulted in some degree of misclassification of vitamin D status; however, given that blood samples were collected prior to conception and laboratory measurement of 25(OH)D was blind to the outcome, we expect this misclassification to be non-differential.”

6. Was sex ratio obtained in a fashion masked to the vitamin D status?

Response: Yes - the vitamin D status as measured for this study was unknown when the sex of the infant was determined by the physician who delivered the baby. Serum was collected before conception (i.e., at the start of the pregnancy attempt) and stored until vitamin D was measured by a separate laboratory from the clinic sites after pregnancy follow-up/delivery was complete. Whether delivering physicians were aware of any clinically-conducted vitamin D assessments before or during pregnancy is unknown to us. The laboratory staff performing vitamin D measurements were completely blinded to all participants (mother-offspring) data, including offspring sex.

7. Could there be reverse causation here? If vitamin D levels are obtained after the sex of the pregnancy is set. I remain somewhat unclear on the timing of vitamin D assessments relative to fetal development.

Response: To clarify, this was a prospective study and vitamin D levels were measured in samples collected and stored before conception (i.e., at the start of the pregnancy attempt). As such, reverse causation is not possible. To further clarify the timing of vitamin D measurements, we have noted throughout the manuscript that the exposure was preconception vitamin D.

8. Page 13, line 336. Are continuous variables included using flexible modeling strategies like splines? Are first-order product terms included. Please provide some details about the IP weights, like mean, min, max, 1 and 99 percentiles.

Response: We appreciate the reviewer’s suggestion to evaluate possible non-linear associations of vitamin D with male live birth. In response, we examined the possibly non-linear relationship between 25(OH)D concentrations and male live birth non-parametrically utilizing restricted cubic spline models and importantly found no strong evidence of non-linearity. Specifically, in separate models, we specified 3, 4, and 5 knots, and evaluated the individual spline term contributions to the model fit and overall test for nonlinearity. The results from these models indicate no departures of linearity (*P*-values for overall tests of non-linearity all >0.48 for models with 3, 4, and 5 knots, respectively). We added the following details regarding this analysis on lines 81-82 and 344-347 in the text: “Results from restricted cubic spline models supported the linear fit of models evaluating continuous 25(OH)D and male live birth (Figure 2)” and “We examined the possibly nonlinear relationship between continuous 25(OH)D and male live birth non-parametrically using restricted cubic spline models (with 3 to 5 knots specified) and evaluated the individual spline term contributions to the model fit and overall test for nonlinearity.”

We have also provided details regarding our IP weights on lines 353-356, as well as in the table below. The same predictors were included in each of the models used to generate the weights: 25(OH)D (continuous), age (continuous), BMI (continuous), number of previous losses (1, 2, or >2), number of previous live births (0, 1, or >1), treatment arm (LDA vs. placebo), marital status (married or unmarried), season of blood draw (spring, summer, fall, winter).

Weight	Mean	Min	Max	1 st percentile	99 th percentile
Loss to follow-up	1.00	0.17	3.40	.322	1.75
Pregnancy	1.01	0.06	4.82	0.19	2.58
Live birth	1.07	0.04	11.66	0.14	4.93

9. Page 15. PLEASE do not state or use a cut point for statistical significance. Report findings (point estimates or bounds) that you believe are important, along with measures of uncertainty, like compatible (aka confidence) intervals or posterior probability distributions. (Perhaps report all P values to 3 decimal places?)

Response: Thank you for your attention to this - we have replaced *P*-values with estimates and confidence intervals throughout the text and tables.

Reviewers' comments:

Reviewer #1 (Remarks to the Author):

The authors have been accommodating in their response and I appreciate their comments. For the most part I feel they have addressed my questions, with just a couple exceptions.

Item #3: The half-life of 25OHD is about two weeks and the authors do not have repeated samples to estimate stability of 25OHD over time. Conception in the second or third cycle is already 60 to 90 days past the measured value and each person is a different distance from their blood measure when they conceive. Thus, fertility is correlated with the misclassification of 25OHD. Women with lower fertility, who take longer to conceive, will be more likely to have misclassified 25OHD levels, which might lead to differential misclassification. As I understand the authors' response, they don't have any data to support that the change is "not substantial". If that's true, it would be better to edit the text to say that as a limitation, this study did not collect data on change in 25OHD over time in this population. It may also make sense to say that misclassification may be differential or non-differential (which can still cause bias).

Item #9: How were values below the LOD treated?

Reviewer #2 (Remarks to the Author):

Review NCOMMS-19-11592.revision: The role of maternal preconception vitamin D status in human offspring sex ratio.

I summarized the reported findings in my prior review. There were no fundamental changes to the manuscript revision.

My Primary Concern with the original paper still remains. I disagree with using the "intention-to-treat" approach to analyze sex-ratio data in this study for several reasons.

(1) The conceptual framework of intent-to-treat pre-supposes a question that would plausibly be addressed by a clinical trial. This is not a question for a clinical trial. One would never limit vitamin D or give vitamin D supplements in order to bring about the birth of a baby of a particular sex, so there is no need to try to design analyses to mimic a trial.

(2) Even if one wanted to do this ITT analysis to try to build a causal argument, estimating sex ratio among babies of women who have not conceived the pregnancy that is supposed to contribute sex data could be far from valid. Presumably estimates are modeled on pregnancies that do occur and not births that do not occur, making the assumption that those outcomes are interchangeable, which is unlikely. The IPW weightings are bound to be imperfect. Unmeasured nutritional factors, early life exposures, behavioral factors, or genetics may all be unbalanced between the women who conceived and the women who did not conceive. The estimates in Table 2 for "all women" (1094 who completed the study of the 1228 enrolled) are stronger than those for pregnancies and births, yet this group included many women who did not have a study pregnancy and even more who did not have a study birth. The authors actually have sex ratio at birth for only 601/1228 (49%) of the women enrolled.

(3) Even if an ITT framework were acceptable and weights were perfect, this sample of women does not approximate a population-based sample useful for generalizing to a broader population. The full sample of study participants, who were themselves selected by two different eligibility criteria, were selected to be planning a pregnancy, have 1 or two prior miscarriages, no more than one prior elective termination, no more than two prior births, no known fertility problems, and regular menstrual cycles. This is not representative of the general population of women giving birth (e.g., in the general population nearly half of pregnancies are unplanned). The sample also has a low overall sex ratio at birth of 0.945 for the 601 babies with data on sex (substantially different from the 1.04-1.05 expected based on vital records for this time period).

(4) The sex ratio at conception and early pregnancy is certainly of interest, but given the problems with this IPW approach, a more direct measure might be possible. Cell-free DNA from the embryo/fetus is found in maternal blood very early in pregnancy, and identification of male DNA may be quite reliable by the mid-1st trimester. For the pregnancies that survived through weeks 7 or 8, this more direct measure could be attempted in this cohort. As I recall, the study collected maternal blood in early pregnancy.

After wondering what a more straight-forward analysis would show, I finally did the calculation for crude sex ratio at birth for the high and low vitamin D groups. The comparison between high and low Vitamin D (RR=1.10, see below) is suggestive, but not compelling. When the data are divided by CRP concentrations, the High CRP group shows a higher RR, but the estimate is well within the confidence intervals of the RR in the Low CRP group. Again, it is interesting but not compelling.

Sample Male N Female N Total N % Males RR (Cis) for Males P value

Overall, low VitD 136 159 295 .46

Overall, high VitD 156 150 306 .51 1.10 (0.94,1.30) 0.23

Low CRP, low VitD 88 92 180 .49

Low CRP, high VitD 111 106 217 .51 1.05 (0.86,1.28) 0.65

High CRP, low VitD 48 67 115 .42

High CRP, high VitD 45 44 89 .51 1.21 (0.90,1.63) 0.21

Other Concerns

1. The text added about the Trivers-Willard hypothesis in the introduction and discussion is sometimes a bit misleading.

Line 30 ("in unfavorable circumstances, females will be less likely to bear sons") suggests this is the direction always expected. The whole hypothesis applies only if the fitness distribution of the two sexes differs substantially and those differences are maintained over generations. When the variance in fitness for males is high compared with females, an increase in females would be expected in unfavorable circumstances. However, if the variance in fitness for females is high relative to males, the opposite is hypothesized.

Line 144: "across mammalian species" implies that a greater variance in fitness for males compared to females is the universal norm for mammals, but that is not the case.

Lines 153-155: Are the references being correctly cited (e.g., the reference for "disasters" refers to the sex of babies born to dioxin-exposed men over a 20-year period; dioxin exposure to women was not associated with baby's sex)? Also, consistency of findings for specific factors can be poor (e.g., the data on sex ratio during wars have been quite variable (James and Valentine, 2014). The data in reference 9 on sex of pregnancy losses is based on karyotype data from 20 euploid losses which seems a bit limited to claim that it "implicates inflammation as a possible biologic pathway."

2. I still don't understand the value of the karyotype data. The timing of the losses is not provided and the selection inherent in the sample of successfully-karyotyped embryos/fetuses means that those data are not reflecting the sex of the population of clinical losses. What exactly do they provide that is valuable for testing the hypothesis?

3. Lines 187-195 were added in response to the prior review, but the arguments are very speculative. The Nepomnaschy study is best viewed as null for sexual dimorphism in hCG during the 1st week after implantation. Furthermore, those data are limited to pregnancies that survived to birth; so, they are selected for survivorship. They do not represent all implanted conceptuses.

4. Lines 239-243: The statement that "1) our results imply that vitamin D sufficiency restores the reduced probability of male live birth among women with elevated inflammation and would not impact the probability of spontaneously conceiving a male or female per se; and, 2) our population estimates would not translate to appreciable change in the probability of male live birth at the individual level." is confusing, given the claims made. In Table 3 the results show a risk difference of 5.9% for the high vitamin D group compared to the low vitamin D (about a 6% elevated risk of having a boy). Doesn't this mean that on average the high vitamin D women would be expected to have a 6% higher probability of having a pregnancy with a male than women in the low vitamin D group? Perhaps the authors just want to remind the reader that even if high vitamin D appears to influence sex ratio with a 6% increase, on average, the probability of having a pregnancy with a male conceptus/embryo/fetus, the probability for any given pregnancy is still either 0 or 1 (ignoring the rare intersex)?

Minor Comments

1. Line 37: The claim of "direct evidence" is overstated; the cited manuscript provides the same sort of IPW, indirect evidence as presented in this current paper.

2. Line 52: Authors may want to cite Jukic et al., 2019 regarding vitamin D and fecundability because it presents additional data and include a review of the prior studies.

3. Line 114: It would be more clear to edit as follows: “. . . RRs for pregnancy with male offspring in the high vitamin D group ranged from 0.71 to 1.77.” (inserting the underlined phrase).
4. Line 208: Unfortunately, a very recent overview of 42 systematic reviews concluded: “RCTs showed no effect of vitamin D supplementation in pregnancy with the exception of one predefined outcome, which had low quality evidence. Credibility of the evidence in this field is compromised by study limitations (in particular, the possibility of confounding among observational studies), inconsistency, imprecision and potential for reporting and publication biases (Bialy et al, 2020).”
5. Line 361: “56” should be “55”, I assume.

Reviewer #4 (Remarks to the Author):

No additional comments.

We thank the Reviewers and the editorial team for their efforts and believe that the changes made to the manuscript have greatly improved the quality and clarity of our paper. The data presented in this version of the manuscript text and tables represents a final clean run of the programs. Please note that in the process of incorporating reviewer requests, the estimates have shifted very slightly, but not meaningfully, from the original version. The final datasets and statistical analysis programs used to generate the results and figures are included in our resubmission. Please also note that to comply with editorial manuscript requirements, we reduced the length of the abstract and included an additional paragraph in the introduction, which describes our overall findings and conclusions.

Reviewers' comments:

Reviewer #1 (Remarks to the Author):

The authors have been accommodating in their response and I appreciate their comments. For the most part I feel they have addressed my questions, with just a couple exceptions.

Response: We thank the reviewer for their positive remarks and helpful review.

Item #3: The half-life of 25OHD is about two weeks and the authors do not have repeated samples to estimate stability of 25OHD over time. Conception in the second or third cycle is already 60 to 90 days past the measured value and each person is a different distance from their blood measure when they conceive. Thus, fertility is correlated with the misclassification of 25OHD. Women with lower fertility, who take longer to conceive, will be more likely to have misclassified 25OHD levels, which might lead to differential misclassification. As I understand the authors' response, they don't have any data to support that the change is "not substantial". If that's true, it would be better to edit the text to say that as a limitation, this study did not collect data on change in 25OHD over time in this population. It may also make sense to say that misclassification may be differential or non-differential (which can still cause bias).

Response: In response to the reviewer, we have added language in the discussion acknowledging potential misclassification (which could be differential or non-differential) of 25(OH)D over time on lines 213-217: “First, only one preconception measurement of 25(OH)D was available in our study population and changes in 25(OH)D concentrations over time may have resulted in some degree of misclassification of vitamin D status at the time of conception, which could be non-differential or differential. Reassuringly, 75% of women conceived in the first 3 months of follow-up and data from a prior study of 68 healthy reproductive-age women suggested that 25(OH)D concentrations measured in the follicular phase of two consecutive cycles did not meaningfully change from cycle 1 to 2. (Harmon et al)”

Reference: Harmon QE, Kissell K, Jukic AMZ, et al. Vitamin D and Reproductive Hormones Across the Menstrual Cycle. *Hum Reprod.* 2020;35(2):413–423. doi:10.1093/humrep/dez283

Item #9: How were values below the LOD treated?

Response: The 21 hsCRP values below the LOD were imputed as $LOD/\sqrt{2}$. We added this information to the text on lines 297: “Values below the LOD were imputed as $LOD/\sqrt{2}$. (Hornung et al)”

Reference: Hornung, R. W. & Reed, L. D. Estimation of Average Concentration in the Presence of Nondetectable Values. *Applied Occupational and Environmental Hygiene* 5, 46-51, doi:10.1080/1047322X.1990.10389587 (1990).

Reviewer #2 (Remarks to the Author):

Review NCOMMS-19-11592.revision: The role of maternal preconception vitamin D status in human offspring sex ratio.

I summarized the reported findings in my prior review. There were no fundamental changes to the manuscript revision. My Primary Concern with the original paper still remains. I disagree with using the “intention-to-treat” approach to analyze sex-ratio data in this study for several reasons.

Response: We thank the reviewer for their efforts on our manuscript.

(1) The conceptual framework of intent-to-treat pre-supposes a question that would plausibly be addressed by a clinical trial. This is not a question for a clinical trial. One would never limit vitamin D or give vitamin D supplements in order to bring about the birth of a baby of a particular sex, so there is no need to try to design analyses to mimic a trial.

Response: The approach of emulating a target trial to frame and refine research questions and identify avoidable sources of bias is well-described in the literature (Hernán et al, Lebreque et al). There are many instances in which conducting a clinical trial to answer a particular research question would not be feasible, ethical, or practical, but by applying trial design principles, even when an analogous trial is not possible, we explicitly tie the analysis to the research questions that we aim to answer. In our approach, we used three different denominators in our risk calculations to answer three distinct questions. In this paper we were primarily interested in estimating the population effect of *preconception* vitamin D on male live birth, which answers the question, “What is the effect of preconception vitamin D sufficiency versus insufficiency on male live birth in a real world setting, where some of the women will not become pregnant or will experience a

pregnancy loss?” Our analysis among the 1,094 women who completed follow-up directly answers this question. In order to get closer to estimating the biological effect, we must account for the factors that led us to be removed from the biological effect (not becoming pregnant or experiencing a pregnancy loss) and answer the following questions: 1) “What is the effect of preconception vitamin D sufficiency versus insufficiency on male live birth, if all women become pregnant?” and, 2) “What is the effect of preconception vitamin D sufficiency versus insufficiency on male live birth, if all women become pregnant and have a live birth?” In our weighted analyses among 803 pregnancies, we get closer to the biological effect by removing the influence of subfertility (not becoming pregnant) on the estimates. In our weighted analyses among 601 live births, we remove the influences of not becoming pregnant and pregnancy losses on the estimates, and therefore estimate the biological effect of preconception vitamin D on male live birth. By using the three-denominator approach, we answer three distinct and important research questions that address both the population effect and the biological effect of preconception vitamin D on male live birth. Importantly, because the timing of our exposure is during the *preconception* period, weights are absolutely required to minimize bias when selecting on pregnancies and live births, for reasons discussed in detail in our response to the Reviewer’s second point below.

In response to the reviewer, we removed the following language from the manuscript on lines 332 and 334 to avoid any potential confusion about this approach: “This approach is analogous to the intent-to-treat analysis of a hypothetical trial in which women are randomly assigned to sufficient or insufficient vitamin D status at preconception and then followed for male live birth for the prespecified duration of the trial.”

We also provided the following clarifications on lines 329-333: “The primary analysis evaluated associations of vitamin D status with the probability of carrying and giving birth to a live-born male among all women who completed follow-up. Risk calculation using this denominator represents the population effect of preconception vitamin D status on male live birth in a real-world setting where among women attempting to conceive, some women go on to achieve pregnancy, while others may experience a pregnancy loss or not achieve pregnancy.

And on lines 361-366: “The live birth-weighted estimates from these analyses describe the association of preconception vitamin D status with male live birth that would have been observed if all women in the study had completed follow-up and had an hCG-detected pregnancy that survived to birth. By limiting the denominator to live births and using inverse-probability weights to account for selection bias imposed by this restriction, we remove the influences of subfertility (not becoming pregnant) and pregnancy loss, and therefore estimate the biological effect of preconception vitamin D status on male live birth.”

References:

Hernán MA, Robins JM. Using Big Data to Emulate a Target Trial When a Randomized Trial Is Not Available. *Am J Epidemiol.* 2016;183(8):758–764. doi:10.1093/aje/kwv254

Labrecque JA, Swanson SA. Target trial emulation: teaching epidemiology and beyond. *Eur J Epidemiol.* 2017;32(6):473–475. doi:10.1007/s10654-017-0293-4

(2) Even if one wanted to do this ITT analysis to try to build a causal argument, estimating sex ratio among babies of women who have not conceived the pregnancy that is supposed to contribute sex data could be far from valid. Presumably estimates are modeled on pregnancies that do occur and not births that do not occur, making the assumption that those outcomes are interchangeable, which is unlikely. The

IPW weightings are bound to be imperfect. Unmeasured nutritional factors, early life exposures, behavioral factors, or genetics may all be unbalanced between the women who conceived and the women who did not conceive. The estimates in Table 2 for “all women” (1094 who completed the study of the 1228 enrolled) are stronger than those for pregnancies and births, yet this group included many women who did not have a study pregnancy and even more who did not have a study birth. The authors actually have sex ratio at birth for only 601/1228 (49%) of the women enrolled.

Response: We would like to reiterate that our analytical approach, which uses three denominators, answers three distinct and valid questions, as described in our response to the reviewer’s point #1 above. The analysis among 1,094 women estimates the population effect (real-world scenario among pregnancy planners in which not all become pregnant and some experience a pregnancy loss), whereas the analysis among 601 live births estimates the biological effect (influences of subfertility and pregnancy loss are removed by restricting to live births and inverse-probability weighting). Again, it is important to emphasize that because we are evaluating a *preconception* exposure, weighting is required to minimize bias when conditioning on pregnancies and live births.

It is not immediately clear to the authors what the reviewer means by “modeled on pregnancies that do occur and not births that do not occur.” We assume the reviewer is suggesting that the only valid analysis is an unweighted analysis restricted to live births only (as they suggest in their point #4 below). However, conditioning on live birth without appropriately weighting (Figure 1) when our question pertains to *preconception* vitamin D status results in either collider stratification bias or overadjustment bias (Schisterman et al 2009, Hernan et al 2004) and will *always* produce biased estimates, which is evidenced by the attenuated risk ratio in the Reviewer’s calculations in their point #4 below.

In Figure 1, we present a simplified directed acyclic graph depicting the relationship between preconception vitamin D and the secondary sex ratio to illustrate why conditioning on live birth without weighting introduces bias. In this causal system, we hypothesize that preconception vitamin D levels may influence a woman’s ability to become pregnant and have a live birth. We also hypothesize that preconception vitamin D may influence the secondary sex ratio (the question we seek to answer in our paper). The secondary sex ratio is observed upon birth and there are likely unmeasured factors (U) that influence both live birth and the secondary sex ratio. In this setting, live birth represents a collider and restricting the analysis to (i.e., conditioning on) live births will result in a biased effect estimate. This DAG follows the classic selection bias structure (Hernan 2004) and inverse probability weights are required to appropriately account for this selection bias. Even if there were no unmeasured factors influencing both live birth and the secondary sex ratio (i.e., U does not exist), conditioning on live birth, as a causal intermediate, would still distort the effect estimates (overadjustment bias).

Figure 1. Simplified directed acyclic graphs (DAG) depicting the relationship of preconception vitamin D on male live birth. U is any unmeasured factor(s) that may influence live birth and the secondary sex ratio. Conditioning on live birth (represented by the black box) opens a backdoor path between U and the outcome, secondary sex ratio, biasing the effect estimate of preconception vitamin D on the secondary sex ratio. This scenario depicts collider stratification bias (Hernan 2004). Weights are required to minimize this bias.

The reviewer is correct, however, that the models used to generate the weights are likely imperfect (mis-specified) to at least some degree, which may result in incomplete adjustment for the selection bias imposed by restricting to/conditioning on live births. Deciding which predictors and their parameterizations to include in the weight models is a bias/variance tradeoff, and a correctly specified model requires that the mean of the weights is ~ 1 , with a relatively small range. (Cole 2009) As demonstrated in our response to the statistical reviewer in the previous round of review (Table 1), our weights appear to be relatively well-behaved according to these criteria. Regardless, in this context, it seems unlikely that the net bias in an analysis restricted to live births without weights would be less than the net bias of the analysis including potentially mis-specified weights.

Table 1. Characteristics of inverse-probability weights

Weight	Mean	Min	Max	1 st percentile	99 th percentile
Loss to follow-up	1.00	0.17	3.40	.322	1.75
Pregnancy	1.01	0.06	4.82	0.19	2.58
Live birth	1.07	0.04	11.66	0.14	4.93

We respectfully request to retain our statistical analyses as they are currently presented. However, we added additional language to acknowledge potential mis-specification of weight models in the text on lines 225-230: “Importantly, the regression models used to generate inverse-probability weights to account for selection bias arising from loss to follow-up and in analyses limited to pregnancies and live birth are subject to potential model misspecification. Necessary conditions of correct model specification require that the mean of the weights approximately equal to 1 and have no extreme outliers. On these bases, the distributions of our estimated weights support correct model specification.”

References:

Stephen R. Cole, Miguel A. Hernán, Constructing Inverse Probability Weights for Marginal Structural Models, *American Journal of Epidemiology*, Volume 168, Issue 6, 15 September 2008, Pages 656–664, <https://doi.org/10.1093/aje/kwn164>

Schisterman EF, Cole SR, Platt RW. Overadjustment bias and unnecessary adjustment in epidemiologic studies. *Epidemiology*. 2009;20(4):488–495. doi:10.1097/EDE.0b013e3181a819a1

Hernán MA, Hernández-Díaz S, Robins JM. A structural approach to selection bias. *Epidemiology*. 2004;15(5):615–625. doi:10.1097/01.ede.0000135174.63482.43

(3) Even if an ITT framework were acceptable and weights were perfect, this sample of women does not approximate a population-based sample useful for generalizing to a broader population. The full sample of study participants, who were themselves selected by two different eligibility criteria, were selected to be planning a pregnancy, have 1 or two prior miscarriages, no more than one prior elective termination, no more than two prior births, no known fertility problems, and regular menstrual cycles. This is not representative of the general population of women giving birth (e.g., in the general population nearly half of pregnancies are unplanned). The sample also has a low overall sex ratio at birth of 0.945 for the 601 babies with data on sex (substantially different from the 1.04-1.05 expected based on vital records for this time period).

Response: Generalizability and transportability are a potential concern in any observational study or trial. We agree with the reviewer that our estimates may not be generalizable to the general population and added language clarifying the target population to whom we are generalizing on lines 240-242: “We therefore anticipate that our findings may be generalizable to similar populations of healthy women planning a pregnancy with comparable levels of inflammation and similar reproductive history.”

The reviewer correctly points out that our study population consisted of pregnancy planners who had 1 or 2 prior miscarriages (a very common pregnancy outcome), no more than one prior elective termination, no more than 2 prior births, no known fertility problems, and regular menstrual cycles. We, of course, agree that these women do not necessarily represent the general population in terms of various lifestyle factors, underlying health status, etc. Our research questions pertain to *preconception* vitamin D, and one of the notable challenges of studying preconception exposures and pregnancy outcomes is that it necessitates recruiting a study sample of women planning a pregnancy, a group who may differ from the general population in terms of underlying health behaviors, socioeconomic status, and other potential factors. These eligibility criteria are crucially important design aspects of our study in that they aid in reducing potential confounding (as discussed in the text on lines 233-240) and minimizing loss to follow-up, therefore improving the internal validity of our study. Furthermore, the inclusion of only women who have 1-2 prior pregnancy losses increases the statistical efficiency of analyses evaluating sex

ratio as an outcome because these women are at higher risk of a biased sex ratio, assuming that a biased sex ratio arises from sex-specific rates of implantation failure or pregnancy loss. This is evident in the relatively low overall sex ratio (0.945) in our study population compared to the general population (1.04-1.05), as the reviewer correctly notes. The low sex ratio in our study population alone, however, is not sufficient to preclude generalizing our findings to the target population explicitly stated: “similar populations of healthy women planning a pregnancy with comparable levels of inflammation and reproductive history.” In order to assert that these findings are not generalizable to this target population, one would need some reason to expect that the association between preconception vitamin D and male live birth would be biologically different in our study sample versus the target population. Though this is a unique population, there is no hypothesized rationale as to why the effect estimate would be expected to be vastly different in a similarly healthy population of women attempting to conceive. As such, it is difficult to agree that this is major concern of the paper.

(4) The sex ratio at conception and early pregnancy is certainly of interest, but given the problems with this IPW approach, a more direct measure might be possible. Cell-free DNA from the embryo/fetus is found in maternal blood very early in pregnancy, and identification of male DNA may be quite reliable by the mid-1st trimester. For the pregnancies that survived through weeks 7 or 8, this more direct measure could be attempted in this cohort. As I recall, the study collected maternal blood in early pregnancy. After wondering what a more straight-forward analysis would show, I finally did the calculation for crude sex ratio at birth for the high and low vitamin D groups. The comparison between high and low Vitamin D (RR=1.10, see below) is suggestive, but not compelling. When the data are divided by CRP concentrations, the High CRP group shows a higher RR, but the estimate is well within the confidence intervals of the RR in the Low CRP group. Again, it is interesting but not compelling.

Sample	Male N	Female N	Total N	% Males	RR (Cis) for Males	P value
Overall, low VitD	136	159	295	.46		
Overall, high VitD	156	150	306	.51	1.10 (0.94,1.30)	0.23

Low CRP, low VitD	88	92	180	.49		
-------------------	----	----	-----	-----	--	--

Low CRP, high VitD	111	106	217	.51	1.05 (0.86,1.28)	0.65
-----	-----	-----	-----	------------------	------

High CRP, low VitD	48	67	115	.42		
----	----	-----	-----	--	--

High CRP, high VitD	45	44	89	.51	1.21 (0.90,1.63)	0.21
----	----	----	-----	------------------	------

Response: To the reviewer’s first point, we agree that an ideal study design might ascertain the primary sex ratio of all conceptions, perhaps using cell-free DNA. This would be an endeavor for future research, but unfortunately is outside the scope of feasibility for the current paper.

To the reviewer’s point about a “more straightforward analysis”, we reiterate our response to their point #2 above. While the reviewer’s crude calculations are mathematically correct, we respectfully contend that the approach is methodologically incorrect in the context of a preconception exposure. Restricting the analysis to 601 live births without implementing weights is not an appropriate analysis because it imposes a selection bias when evaluating preconception exposures on birth outcomes, as illustrated above. Of note, even when the incorrect approach (restricted to live births without weighting) is used, the estimates still show a positive association of vitamin D and male live birth, just with reduced efficiency owing to reduced sample size.

Other Concerns

1. The text added about the Trivers-Willard hypothesis in the introduction and discussion is sometimes a bit misleading. Line 30 (“in unfavorable circumstances, females will be less likely to bear sons”) suggests this is the direction always expected. The whole hypothesis applies only if the fitness distribution of the

two sexes differs substantially and those differences are maintained over generations. When the variance in fitness for males is high compared with females, an increase in females would be expected in unfavorable circumstances. However, if the variance in fitness for females is high relative to males, the opposite is hypothesized.

Response: We softened our language regarding the Trivers-Willard hypothesis on lines 3 and 30.

Line 3: “Evolutionary theory suggests that the offspring sex ratio of some animal species may shift in response to maternal health status and environmental conditions, such that under some unfavorable conditions, females may be less likely to bear sons.”

Line 30: “According to evolutionary theory, the ability of the offspring sex ratio to shift in some animal species to adjust in response to maternal health status and resource availability confers a natural selection advantage, and in some unfavorable circumstances, females may be less likely to bear sons.¹”

Line 146: “across mammalian species” implies that a greater variance in fitness for males compared to females is the universal norm for mammals, but that is not the case.

Response: We edited this sentence accordingly on line 142: “In some, but not all, mammalian species, males exhibit greater variance in terms of lifetime reproductive success, with much of this variance explained by specific advantageous physical and social characteristics.²²”

Lines 153-155: Are the references being correctly cited (e.g., the reference for “disasters” refers to the sex of babies born to dioxin-exposed men over a 20-year period; dioxin exposure to women was not associated with baby’s sex)? Also, consistency of findings for specific factors can be poor (e.g., the data on sex ratio during wars have been quite variable (James and Valentine, 2014). The data in reference 9 on sex of pregnancy losses is based on karyotype data from 20 euploid losses which seems a bit limited to claim that it “implicates inflammation as a possible biologic pathway.”

Response: In response to the reviewer, we replaced this on lines 26 and 152. We also revised the sentence on lines 150-151: “Though the consistency of studies of sex ratio varies widely, many studies report diminished sex ratios in human populations following disasters¹², wars²⁴, and times of economic depression²⁵, and recent randomized clinical trial data implicated inflammation as a possible biologic pathway.⁹”

To clarify, the paper by Radin et al (reference #9) refers to an unweighted analysis of a randomized controlled trial examining the effect of preconception-initiated daily LDA on the probability of male live birth. The authors also included weighted analyses among pregnancies and live births, as well as karyotype data in a secondary analysis, as we did in our paper. In the Radin et al paper (reference #9), the ITT unweighted analyses, weighted analyses, and the secondary analyses including karyotype data on pregnancy losses each showed an effect of aspirin treatment (an anti-inflammatory drug) on improving the probability of male live birth, with the strongest effects noted for women with elevated preconception hsCRP. We therefore request to retain this information regarding reference 9 and its support of inflammation as a possible pathway to an altered sex ratio.

2. I still don’t understand the value of the karyotype data. The timing of the losses is not provided and the selection inherent in the sample of successfully-karyotyped embryos/fetuses means that those data are not reflecting the sex of the population of clinical losses. What exactly do they provide that is valuable for testing the hypothesis?

Response: The median gestational age of karyotyped losses was 8 weeks (IQR: 7-10 weeks, range 2-20 weeks); we added this information to the manuscript on lines 308-309. The goal of incorporating the karyotype data was to estimate the association of preconception vitamin D and sex ratio shortly after implantation, using all data available to us, albeit with missingness. While the estimates from these secondary analyses support the primary findings of a positive association of preconception vitamin D on male live birth, we took great care to evaluate the potential impact of missingness of fetal sex of these pregnancy losses on the estimates in a sensitivity analysis. Indeed, the sensitivity analysis suggests that the estimates from the analyses including karyotype data may be explained by missing data and we acknowledged this limitation in the manuscript. Importantly, missingness of fetal sex among pregnancy losses would not explain our primary findings, that preconception vitamin D is positively associated with male live birth.

3. Lines 187-195 were added in response to the prior review, but the arguments are very speculative. The Nepomnaschy study is best viewed as null for sexual dimorphism in hCG during the 1st week after implantation. Furthermore, those data are limited to pregnancies that survived to birth; so, they are selected for survivorship. They do not represent all implanted conceptuses.

Response: We added language acknowledging uncertainty regarding this mechanism on lines 192-193: “Though the precise mechanism is unclear, it is possible that lower hCG secretion by male preimplantation embryos may confer higher susceptibility to preimplantation failure among women with elevated inflammation; adequate 25(OH)D levels may provide some remediation, thereby increasing the likelihood of survival and eventual implantation of male embryos.”

4. Lines 239-243: The statement that “1) our results imply that vitamin D sufficiency restores the reduced probability of male live birth among women with elevated inflammation and would not impact the probability of spontaneously conceiving a male or female per se; and, 2) our population estimates would not translate to appreciable change in the probability of male live birth at the individual level.” is confusing, given the claims made. In Table 3 the results show a risk difference of 5.9% for the high vitamin D group compared to the low vitamin D (about a 6% elevated risk of having a boy). Doesn’t this mean that on average the high vitamin D women would be expected to have a 6% higher probability of having a pregnancy with a male than women in the low vitamin D group? Perhaps the authors just want to remind the reader that even if high vitamin D appears to influence sex ratio with a 6% increase, on average, the probability of having a pregnancy with a male conceptus/embryo/fetus, the probability for any given pregnancy is still either 0 or 1 (ignoring the rare intersex)?

Response: We revised this statement on lines 245-252: “Importantly, these data should not be interpreted as a method to influence infant sex, given that our results imply that vitamin D sufficiency restores the reduced probability of male live birth among women with elevated inflammation and would not impact the probability of spontaneously conceiving a male or female per se.

To further clarify, we interpret our data to indicate is that having sufficient versus insufficient preconception vitamin D restores the probability of a male conceptus surviving through birth. The probability of spontaneously conceiving a male (or female) embryo is ~50% (not 0 or 1) and having sufficient versus insufficient vitamin D levels appears to, on average, increase the likelihood that male conceptuses survive to birth when low-grade inflammation is present.

Minor Comments

1. Line 37: The claim of “direct evidence” is overstated; the cited manuscript provides the same sort of IPW, indirect evidence as presented in this current paper.

Response: We softened our language on line 25: “...suggesting that this phenomenon may also occur in humans.” Of note, this reference does in fact refer to the unweighted analysis evaluating the effect of preconception-initiated daily LDA versus placebo on male live birth in a randomized controlled trial. Inverse-probability weighted analyses among pregnancies and live births (as we did in our paper) were also included and supported the findings of the primary unweighted ITT analyses.

2. Line 52: Authors may want to cite Jukic et al., 2019 regarding vitamin D and fecundability because it presents additional data and include a review of the prior studies.

Response: Thank you for pointing out this new reference (now appears as reference #23). We made the following revisions to the text:

Lines 40-42: “Preconception 25-hydroxyvitamin D concentrations were similarly positively associated with fecundability in two other prospective studies of healthy pregnancy planners,^{22,23}
as well as in couples seeking fertility treatment.^{24,25}”

Lines 163-166: “These findings are in agreement with two other prospective studies, which also reported lower probability of pregnancy among healthy women who were vitamin D insufficient or deficient (25(OH)D <12 ng/mL).^{22,23}”

3. Line 114: It would be more clear to edit as follows: “. . . RRs for pregnancy with male offspring in the high vitamin D group ranged from 0.71 to 1.77.” (inserting the underlined phrase).

Response: Thank you – we made these edits on line 111: “RRs for pregnancy with male offspring comparing the vitamin D sufficient versus insufficient groups....”

4. Line 208: Unfortunately, a very recent overview of 42 systematic reviews concluded: “RCTs showed no effect of vitamin D supplementation in pregnancy with the exception of one predefined outcome, which had low quality evidence. Credibility of the evidence in this field is compromised by study limitations (in particular, the possibility of confounding among observational studies), inconsistency, imprecision and potential for reporting and publication biases (Bialy et al, 2020).”

Response: In response to the reviewer, we edited the sentence on line 208: “Serum 25(OH)D concentrations, though not vitamin D supplementation,³⁹ have been associated with a lower overall risk of these conditions,⁴⁰ and some also reported effect modification by fetal sex, with stronger benefits among males.⁴¹”

Of note, vitamin D supplementation trials are frequently discrepant with observational analyses of vitamin D status across many health outcomes, including osteoporosis. Reasons for these discrepancies have been discussed widely and are not solely attributable to residual confounding among observational studies. In particular, vitamin D supplementation in a trial will not be effective if participants have sufficient vitamin D status at enrollment. Furthermore, vitamin D is obtained primarily from sunlight, with a much smaller proportion coming from diet or supplements. All or some of these issues can contribute to inconsistent findings between RCTs and observational studies.

5. Line 361: “56” should be “55”, I assume.

Response: The number of karyotyped pregnancy losses was in fact 56. Please see lines 307-308: “55 tests determined fetal sex, whereas 29 were not determined due to testing failure. No genetic analysis was available for one phenotypic male (gestational age = 15 weeks).”

Reviewer #4 (Remarks to the Author):

No additional comments.

Response: We thank the reviewer for their helpful remarks.

REVIEWER COMMENTS

Reviewer #1 (Remarks to the Author):

I have no further comments. Thank you!

Reviewer #2 (Remarks to the Author):

I still have concerns about this manuscript.

1. Regarding their response to item 1 in my prior review: I am still worried about their response to my question about their conceptual framework. They reiterate: "In this paper we were primarily interested in estimating the population effect of preconception vitamin D on male live birth, which answers the question, 'What is the effect of preconception vitamin D sufficiency versus insufficiency on male live birth in a real world setting, where some of the women will not become pregnant or will experience a pregnancy loss?' "

However, though they measured 25(OH)D before conception, they are using it as a surrogate for vitamin D at time of conception. Their interest is in its effect at the time of sex determination, as is apparent from their current response to reviewer 1 (item #3) regarding the problem with measurement error of Vitamin D. There they argue that it is likely to be relatively stable over time, and that the majority of women conceived during their first 3 cycles. However, for some women the blood sample is taken 6 or more months before conception, and even when it is only a month or two before, the concentrations can change (sun exposure differences, diet differences, etc.). For women who were using estrogen containing contraceptives before attempt pregnancy, 25(OH)D concentrations are likely to be dropping because both exogenous and endogenous estrogens are associated with higher circulating 25(OH)D (Harmon et al. 2016; Harmon et al. 2020). Also, supplement use can change over time, as couples who do not get pregnant as rapidly as they would like, are likely to change behaviors. They cite Harmon et al., 2020 to support the stability of vitamin D across menstrual cycles but this was based on analyses of data from a sample of women with regular cycles who had not recently used hormonal contraception, were not taking supplements, and were followed for only two cycles.

Though they gain power by starting with the sample of all women who completed the study, I see no conceptual need to start with that group, which includes women who never became pregnant. This analysis is not part of the trial, no longer an intention to treat scenario; it is a secondary analysis of trial data that was not part of the original goals of the trial.

They argue that their IPW weighting adequately provides appropriate weightings. The fact that the mean weight for each set of "missings" is at or close to 1:00, a necessary condition for statistical validity, is good. However, statistical validity does not necessarily mean it is sufficient for valid weight prediction. For example, they include aspirin treatment (the primary question addressed by the trial) as a factor in the weighting for each of the samples (study completion, becoming pregnant, and having a live birth), but they have previously reported that the aspirin treatment was associated with attaining a live birth only in one of their enrollment groups (the original recruitment stratum, not the expanded stratum that allowed for 1-2 losses at any time in the past). Thus, it is the interaction of the two variables (aspirin-treatment group and recruitment stratum) that is predictive of live birth (Levine et al., 2019). The primary findings from the trial report no difference in pregnancy loss by treatment in the expanded recruitment group, only in the original recruitment group (Schisterman et al., 2014). Why was recruitment stratum not included as a variable in the prediction model for developing weights? Am I incorrect that adding extra variables to a prediction model (not a causal model) does not lead to overadjustment bias? This whole issue is concerning given the data the authors provided in their prior rebuttal showing that the sex difference they report was only seen in one of the recruitment groups, the group with a single loss within the last year:

"we stratified our main analyses by the original (1 recent prior loss) and expanded (1-2 prior losses at any gestational age and any time in the past) eligibility criteria for the trial. In doing so, we found that the associations were more pronounced in the original stratum (adjusted RR = 1.49, 95% CI = 1.12-1.96) compared to the expanded stratum (adjusted RR = 1.05, 95% CI = 0.78-1.40). Reasons for differences in findings among women across strata are unclear. However,

compared to women in the expanded stratum, women in the original stratum had slightly lower BMI (25.8 vs. 26.8 kg/m²), lower hsCRP (2.6 vs. 3.2 ng/mL), and slightly lower 25(OH)D levels (30.3 vs. 31.2 ng/mL). The differences in characteristics between these two groups, namely BMI and hsCRP, suggest that the influence of vitamin D on male live birth may be stronger among women with non-obesity-related inflammation. We also postulate that a larger proportion of women in the stratum of 2 prior losses would ultimately go on to be characterized as having recurrent pregnancy loss (>2 prior losses) and may therefore have some different underlying pathology. Importantly, we believe that the stronger findings among women in the original stratum lend support to the generalizability of our findings, in that these women are more representative of the general population of women attempting pregnancy."

Wouldn't it provide valuable information to a reader to include the differential effects by recruitment group, as was done when reporting trial results?

Also, I'm not sure their last statement in their quote above is correct. I haven't seen the data supporting it, but neither have I seen data suggesting it is incorrect. I think that women with losses are now encouraged to go ahead and try again as soon as they're ready, while in the past I think there were some who encouraged "taking a rest" before trying again. Regardless, they expanded the recruitment presumably because it was taking so long to enroll with just the original enrollment criteria.]

2. I'm still unclear about the value of the karyotype data. They note in their response that "The goal of incorporating the karyotype data was to estimate the association of preconception vitamin D and sex ratio shortly after implantation." Of course, that is not possible with karyotype data.

3. Sex-ratio differences rarely replicate. Given the lack of consistency of results between recruitment subsets in this cohort, and given that the only statistically-significant findings in Table 2 are those that rely on IPW weightings to include women who did not conceive, perhaps it would be best to present the results as hypothesis-generating findings.

Reviewer #1 (Remarks to the Author):

I have no further comments. Thank you!

Response: Thank you very much for your constructive and thoughtful review.

Reviewer #2 (Remarks to the Author):

I still have concerns about this manuscript.

1. Regarding their response to item 1 in my prior review: I am still worried about their response to my question about their conceptual framework. They reiterate: “In this paper we were primarily interested in estimating the population effect of preconception vitamin D on male live birth, which answers the question, ‘What is the effect of preconception vitamin D sufficiency versus insufficiency on male live birth in a real world setting, where some of the women will not become pregnant or will experience a pregnancy loss?’ ” However, though they measured 25(OH)D before conception, they are using it as a surrogate for vitamin D at time of conception. Their interest is in its effect at the time of sex determination, as is apparent from their current response to reviewer 1 (item #3) regarding the problem with measurement error of Vitamin D. There they argue that it is likely to be relatively stable over time, and that the majority of women conceived during their first 3 cycles. However, for some women the blood sample is taken 6 or more months before conception, and even when it is only a month or two before, the concentrations can change (sun exposure differences, diet differences, etc.). For women who were using estrogen containing contraceptives before attempt pregnancy, 25(OH)D concentrations are likely to be dropping because both exogenous and endogenous estrogens are associated with higher circulating 25(OH)D (Harmon et al. 2016; Harmon et al. 2020). Also, supplement use can change over time, as couples who do not get pregnant as rapidly as they would like, are likely to change behaviors. They cite Harmon et al., 2020 to support the stability of vitamin D across menstrual cycles but this was based on analyses of data from a sample of women with regular cycles who had not recently used hormonal contraception, were not taking supplements, and were followed for only two cycles.

Response: Thank you for this comment. To clarify, in our analyses, we assume a steady state of vitamin D, where on average, women are generally exposed to similar levels over time throughout the reproductive process. We do not intend to use it as a surrogate for vitamin D at the time of conception, nor is our interest in the effect of preconception vitamin D at the time of sex determination. We believe that this “steady state” assumption is reasonable based on the following:

1) While we agree with the Reviewer that positive associations have been observed between vitamin D and endogenous/exogenous estrogens in cross-sectional studies (Harmon 2016, Harmon 2020), it is unclear the extent to which 25(OH)D levels change after stopping contraceptives, as previous studies did not explicitly evaluate this question. Importantly, only 12% of women in our study population reported using any hormonal contraceptives in the 3 months prior to enrollment (when the blood sample was drawn). The median time between women stopping hormonal contraceptives and the blood draw was 41 months (range = 0-235 months).

2) Other studies have shown that 25(OH)D levels tend to be very stable over time (from 1 to 14 years) (McKibben, 2016; van Schoor, 2014; Jorde, 2010), with some seasonal variability which

we accounted for in this analysis (see models additionally adjusted for season on lines 118-120; RR = 1.26, 95% CI = 1.03, 1.54).

3) The maximum number of preconception months that a woman would have been followed is 6 (not more), per the study protocol. Thus, the maximum amount of time between 25(OH)D measurement and conception would be ~5.5 months, not six months or more, as the Reviewer suggests.

Though biologically plausible, it seems quite unlikely that a substantial proportion of women in our study population would have drastic changes in 25(OH)D levels over the study period based on the items above. More importantly, even if some proportion of women did experience moderate changes in 25(OH)D during the study period, it is difficult to imagine how these changes (measurement error) would be differentially related to the outcome, male live birth, such that it would induce a spurious association. To specifically address the Reviewer’s concern regarding this possibility, however, we conducted analyses limited to conceptions occurring within the first 2 cycles of follow-up (conceptions that occurred ~2 to 6 weeks after 25(OH)D measurement). In the select group of women who conceived in the first 2 cycles following 25(OH)D measurement, estimates were very similar (and even slightly stronger) compared to the overall pregnancy estimates.

		Pregnancies detected in cycle 1+2 (N=430)	
Vitamin D	N (male LB)	Unadjusted	Adjusted
<30 ng/mL	79	1	1
>30 ng/mL	97	1.27 (0.99, 1.63)	1.24 (0.97, 1.59)

We made the following revisions to the discussion to address these points on lines 215-225:

“First, we were primarily interested in the usual rather than acute (i.e., conception cycle) effects of vitamin D on male live birth as our study did not include multiple measurements of preconception 25(OH)D with male live birth. Changes in 25(OH)D concentrations over time between blood draw and conception (~2 weeks to 5.5 months) may have resulted in some degree of misclassification of vitamin D status, which could be non-differential or differential. Reassuringly, 75% of women in the EAGeR trial conceived in the first 3 months of follow-up and data from a prior study of 68 healthy reproductive-age women suggest that 25(OH)D concentrations measured in the follicular phase of two consecutive cycles did not meaningfully change from cycle 1 to 2.⁴³ Moreover, in sensitivity analyses limited to conceptions occurring in the first 2 cycles of follow-up (~2 to 6 weeks after blood draw), estimates for male live birth comparing vitamin D sufficiency versus insufficiency were consistent with the overall analysis (among 430 pregnancies RR = 1.24, 95% CI = 0.97, 1.59).:

References:

Harmon QE, Kissell K, Jukic AMZ, et al. Vitamin D and Reproductive Hormones Across the Menstrual Cycle. *Hum Reprod.* 2020;35(2):413-423. doi:10.1093/humrep/dez283

Harmon QE, Umbach DM, Baird DD. Use of Estrogen-Containing Contraception Is Associated With Increased Concentrations of 25-Hydroxy Vitamin D. *J Clin Endocrinol Metab.* 2016;101(9):3370-3377. doi:10.1210/jc.2016-1658

McKibben RA, Zhao D, Lutsey PL, et al. Factors Associated With Change in 25-Hydroxyvitamin D Levels Over Longitudinal Follow-Up in the ARIC Study. *J Clin Endocrinol Metab.* 2016;101(1):33-43. doi:10.1210/jc.2015-1711

van Schoor NM, Knol DL, Deeg DJ, Peters FP, Heijboer AC, Lips P. Longitudinal changes and seasonal variations in serum 25-hydroxyvitamin D levels in different age groups: results of the Longitudinal Aging Study Amsterdam. *Osteoporos Int.* 2014;25(5):1483-1491. doi:10.1007/s00198-014-2651-3

Jorde R, Sneve M, Hutchinson M, Emaus N, Figenschau Y, Grimnes G. Tracking of serum 25-hydroxyvitamin D levels during 14 years in a population-based study and during 12 months in an intervention study. *Am J Epidemiol.* 2010;171(8):903-908. doi:10.1093/aje/kwq005

Though they gain power by starting with the sample of all women who completed the study, I see no conceptual need to start with that group, which includes women who never became pregnant. This analysis is not part of the trial, no longer an intention to treat scenario; it is a secondary analysis of trial data that was not part of the original goals of the trial.

Response: Thank you for this comment. Our analytic approach, which uses three denominators to answer three distinct and valid research questions, seems to be an intractable point of philosophical disagreement. Please note that we have conducted analyses among pregnancies and live births, exactly as the Reviewer has requested, and these results appear in the main results table (Table 2). In addition, from the outset of our paper, we were also interested in the total effect of preconception vitamin D on male live birth, necessitating our analytic approach of using the denominator of “all women who completed follow-up” (n = 1,094). Using different denominators in risk calculations to answer different questions, including the denominator of “all women”, is considered the “gold standard” modern approach for evaluating preconception exposures and birth outcomes, as recently described by the world-renowned biostatistician and epidemiologist, Miguel Hernan (Chiu and Hernan, 2020). We hope our that our detailed explanation to the Reviewer below demonstrates why answering *each* of these three questions is important. Please note that we added more discussion of the limitations of our study, added/removed analyses, and tempered our conclusions, in an effort to be as responsive as possible to the Reviewer. Finally, the soundness of our statistical approach to answer each of our research questions has been thoroughly reviewed and deemed acceptable by an expert statistical reviewer (Reviewer #4) as well as two additional reviewers.

Next, we describe in detail the potential for vitamin D to influence multiple steps in the reproductive process and the specific questions of interest addressed in the paper. In the Figure below, we illustrate the selection process that occurs from preconception to live birth to highlight the relevance of each of the questions addressed in this analysis. On the left, we start with the full population of women attempting pregnancy (**A**). Moving to the right, as women attempt pregnancy, some will have an egg that is fertilized, resulting in a preimplantation embryo that is male or female (sex has been determined) (**B**). Moving further right, some of these embryos will successfully implant in the uterus, at which point, hCG becomes detectable and the woman will

produce a positive pregnancy test (C). Importantly, up to ~30% of fertilized eggs do not successfully implant; thus, the implanted embryos (pregnancies detected by urine hCG test) represent a highly selected population. Moving even further to the right, the pregnancy continues through gestation; during this time, some pregnancies will end in pregnancy loss (~30% of hCG-detected pregnancies) or stillbirth (~1% of hCG-detected pregnancies). Those who are live born (avoid pregnancy loss and stillbirth) represent an even more highly selected population (D).

It is reasonable to assume that under normal conditions, among the unobservable preimplantation embryos (B), roughly half are male, and half are female (sex ratio at fertilization ~ 1.00). The sex ratio among the hCG-detected pregnancies (C) may then be perturbed if implantation failure is more frequent among either the male or female preimplantation embryos. As gestation continues, the sex ratio may become further perturbed if pregnancy losses or stillbirths are more common among pregnancies with either a male fetus or a female fetus, resulting in an altered sex ratio at birth. Vitamin D levels during the preconception period have the potential to influence each of these selection steps through its immune modulating effects and other mechanisms, and as such it is important to consider the role of vitamin D at each step and account for this in the analytic approach.

In our first research question of interest, “What is the influence of preconception vitamin D on male live birth in a real-world setting, where some women will not conceive (i.e., have a hCG-detected pregnancy), and some will not have a live birth?”, we are agnostic (i.e., make no assumptions) to whether vitamin D influences the likelihood of male live birth by producing more male preimplantation embryos (B), improving their likelihood of successfully implanting (by reducing implantation failure) (C), or reducing their likelihood of pregnancy loss or still birth. In our calculations among 1,094 women, we simply estimate the *total effect* of preconception vitamin D on male live birth *through all of these processes combined*. Consider the corresponding (hypothetical) clinical scenario: women contemplating pregnancy visit their physician, have 25(OH)D measured, and then attempt pregnancy, not knowing when or if they will go on to

become pregnant or have a live birth in the next six months. Referring to our estimates from the full sample (n=1,094), our results can be interpreted as the likelihood of having a male live birth based on current 25(OH)D levels, regardless of whether she will have an hCG-detected pregnancy in the next 6 months or whether the pregnancy will survive to birth. In other words, these estimates are agnostic to whether vitamin D increases male live birth by increasing the fertilization of male embryos, increasing their likelihood of implantation, or reducing their likelihood of loss – it describes the *total effect* of preconception vitamin D on male live birth *through all of these processes combined*. This is an important question, as in this preconception period, women with insufficient vitamin D would still have an opportunity to increase their vitamin D to improve their likelihood of a male embryo properly implanting (becoming an hCG-detected pregnancy) and surviving to birth, making this information both relevant and potentially useful.

Our second research question of interest, “What is the influence of preconception vitamin D on male live birth among those who successfully conceived?”, requires us to change the denominator to 803 hCG-detected pregnancies (while employing weights to minimize selection bias arising from this restriction). In doing so, we are evaluating the influence of vitamin D on male live birth, while removing the potential influence of vitamin D on achieving hCG-detected pregnancy (**C**). Because many preimplantation embryos do not successfully implant (~30%), changing the denominator to the 803 pregnancies removes any influence that preconception vitamin D may have on male live birth through producing more male preimplantation embryos (**B**) and through helping them successfully implant as hCG-detected pregnancies (**C**). Though biologically interesting, the potential practical utility of answering this question is less clear compared to question 1. Consider the hypothetical corresponding scenario: A woman has a preconception visit with her physician, has her 25(OH)D levels measured, and wants to know the likelihood that she will have a male live birth based on her current preconception 25(OH)D status. At that preconception visit, neither she nor her doctor can tell what the future holds - whether she will go on to 1) conceive in the next six months, and 2) carry the pregnancy to a live birth. Referring to the estimates from our analyses among the 803 women who became pregnant, the doctor tells her, “Based on your vitamin D status at this preconception visit, this is the likelihood of a male live birth, but only if conception in the next six months is absolutely guaranteed.” As these estimates pertaining to a preconception measurement are *conditional on guaranteed future hCG-detected pregnancy* (which is not realistic because there are currently no interventions available to ensure pregnancy), their relevance and usefulness to the woman in the preconception period is less clear (though are certainly of biological interest). If she does go on to conceive two months later (for example), this estimate may become more relevant, but the opportunity to alter her *preconception* vitamin D status to improve her likelihood of live birth after producing a male embryo has already passed and the information is no longer potentially useful. Please note that this estimate appears in our main results table (Table 2).

Our third research question, “What is the influence of preconception vitamin D on male live birth among live births?”, requires us to change the denominator to 601 live births (while employing inverse-probability weights to minimize selection bias arising from this restriction). In doing so, we are evaluating the influence of vitamin D on male live birth, while removing the potential influence of vitamin D on both achieving pregnancy and maintaining that pregnancy to live birth (**D**). Changing the denominator to 601 live births takes away the influence of preconception vitamin D may have on male live birth through producing more male preimplantation embryos (**B**), through helping them successfully implant as hCG-detected pregnancies (**C**), and through

helping them avoid pregnancy loss or still birth. While certainly interesting, the potential practical utility of answering this question is somewhat less clear compared to our first research question. The estimates among the live births describe the likelihood of male live birth associated with preconception vitamin D status, *conditional on guaranteed future hCG-detected pregnancy and survival to live birth*. While biologically interesting, these estimates are not particularly useful to the woman in the preconception period, because there are currently no interventions available to ensure both implantation and survival to birth. Regardless, we agree with the Reviewer that this question is of biologic interest and as such, the estimates corresponding to this research question appear in Table 2, exactly as the Reviewer has requested.

The need for using different denominators, including the denominator of “all women attempting to conceive”, in the context of a preconception exposure and birth outcome to answer different meaningful research questions is considered the “gold standard” approach, as discussed at length by others. (Chiu and Hernan, 2020) Importantly, we present results from each of these analyses in our main results table (Table 2) for the reader. We would also like to highlight that the point estimates from each of these three analyses are quite consistent, with expected losses of efficiency (wider confidence intervals) due to smaller sample sizes in analyses restricted to pregnancies and live births. To interpret the estimates among all women who completed follow-up (RR = 1.26; 95% CI = 1.03, 1.53) as evidence of an association, but the estimates among pregnancies (RR = 1.18; 95% CI = 0.97, 1.44) and live births (RR = 1.18; 95% CI = 0.94, 1.48) as “null” or “no association” simply because of reduced efficiency would be incorrect. The pitfalls of such overreliance on significance testing have been well-described across scientific disciplines. (Amrhein et al, 2019)

To summarize, our analytic approach using three denominators was to answer specific research questions. Given our interest in understanding the role of *preconception* vitamin D status, our analysis approach necessarily follows. By changing the denominator to pregnancies and live births (which are also presented in Table 2), the research question addressed in the analyses fundamentally changes. Moreover, restricting to pregnancies and live births introduces selection bias that must be corrected (to the extent that it can be) through use of inverse-probability weights. Though the Reviewer may philosophically disagree about *which* question is most important, *only* presenting analyses among live births simply would not address important and meaningful questions that we intended to answer in this paper from the outset.

References:

Chiu, Yu-Han et al. The Effect of Prenatal Treatments on Offspring Events in the Presence of Competing Events, *Epidemiology*: September 2020 - Volume 31 - Issue 5 - p 636-643 doi: 10.1097/EDE.0000000000001222

Amrhein V, Greenland S, McShane B. Scientists rise up against statistical significance. *Nature*. 2019 Mar;567(7748):305-307. doi: 10.1038/d41586-019-00857-9. PMID: 30894741.

They argue that their IPW weighting adequately provides appropriate weightings. The fact that the mean weight for each set of “missings” is at or close to 1:00, a necessary condition for statistical validity, is good. However, statistical validity does not necessarily mean it is sufficient for valid weight prediction. For example, they include aspirin treatment (the primary question addressed by the trial) as a factor in the

weighting for each of the samples (study completion, becoming pregnant, and having a live birth), but they have previously reported that the aspirin treatment was associated with attaining a live birth only in one of their enrollment groups (the original recruitment stratum, not the expanded stratum that allowed for 1-2 losses at any time in the past). Thus, it is the interaction of the two variables (aspirin-treatment group and recruitment stratum) that is predictive of live birth (Levine et al., 2019). The primary findings from the trial report no difference in pregnancy loss by treatment in the expanded recruitment group, only in the original recruitment group (Schisterman et al., 2014). Why was recruitment stratum not included as a variable in the prediction model for developing weights? Am I incorrect that adding extra variables to a prediction model (not a causal model) does not lead to overadjustment bias?

Response: Thank you for the suggestion to include an interaction term (eligibility strata×treatment group) in our live birth inverse-probability weight models. We added this interaction term and re-ran the analyses. Estimates from these updated models were virtually identical to the original analyses (among 601 live births RR = 1.18; 95% CI = 0.94, 1.48) and did not change the interpretation of the data in any meaningful way.

We also added the following text to the manuscript to acknowledge the Reviewer's concerns regarding inverse-probability weight model misspecification on lines 237-239:

“On these bases, the distributions of our estimated weights support correct model specification; however, we acknowledge that complete elimination of selection bias is unlikely, as unknown or unmeasured common causes of selection factors (i.e., loss to follow-up, pregnancy, and live birth) and male live birth may exist.”

We would like to emphasize that adding additional variables to a prediction model for developing weights will *not* uniformly lead to better prediction and less biased risk estimates. The inclusion of additional variables and parameters in the IPW model can *increase* net bias by introducing random positivity violations, especially in the context of a modest sample size (Cole et al., 2008). Our decisions regarding the predictors included in creating the IPW weights have been informed by the desire to balance competing concerns for bias.

References:

Cole SR, Hernán MA. Constructing inverse probability weights for marginal structural models. *Am J Epidemiol.* 2008;168(6):656-664. doi:10.1093/aje/kwn164

This whole issue is concerning given the data the authors provided in their prior rebuttal showing that the sex difference they report was only seen in one of the recruitment groups, the group with a single loss within the last year: “we stratified our main analyses by the original (1 recent prior loss) and expanded (1-2 prior losses at any gestational age and any time in the past) eligibility criteria for the trial. In doing so, we found that the associations were more pronounced in the original stratum (adjusted RR = 1.49, 95% CI = 1.12-1.96) compared to the expanded stratum (adjusted RR = 1.05, 95% CI = 0.78-1.40). Reasons for differences in findings among women across strata are unclear. However, compared to women in the expanded stratum, women in the original stratum had slightly lower BMI (25.8 vs. 26.8 kg/m²), lower hsCRP (2.6 vs. 3.2 ng/mL), and slightly lower 25(OH)D levels (30.3 vs. 31.2 ng/mL). The differences in characteristics between these two groups, namely BMI and hsCRP, suggest that the influence of vitamin D on male live birth may be stronger among women with non-obesity-related inflammation. We also postulate that a larger proportion of women in the stratum of 2 prior losses would ultimately go on to be characterized as having recurrent pregnancy loss (>2 prior losses) and may therefore have some different

underlying pathology. Importantly, we believe that the stronger findings among women in the original stratum lend support to the generalizability of our findings, in that these women are more representative of the general population of women attempting pregnancy.”

Wouldn't it provide valuable information to a reader to include the differential effects by recruitment group, as was done when reporting trial results?

Response: Thank you for this suggestion. These results were not originally included in our paper because there was no *a priori* hypothesis regarding effect modification by eligibility strata. To address the Reviewer's concern, we have now added results stratified by eligibility criteria in Supplemental Table 5 and on lines 122-125 in the results section. We also included a brief discussion of these findings to the paper on lines 184-187:

Lines 122-126: “In post-hoc analyses stratified by eligibility criteria, estimates describing the association of vitamin D status and male live birth were pronounced in the original stratum among women with a single recent loss (RR = 1.48; 95% CI = 1.12, 1.96) compared to the women in the expanded stratum with 1 or 2 prior losses at any time in the past and at any gestational age (RR= 1.06; 95% CI = 0.79, 1.41)”

Lines 180-183: “Moreover, associations were particularly observed among women with a single recent loss, who tended to have lower adiposity on average, which suggests that the influence of vitamin D on male live birth may be stronger among women with non-obesity-related inflammation.”

Supplementary Table 5. Adjusted relative risks (RR) and 95% confidence intervals (CIs) for vitamin D and male live birth, stratified by eligibility criteria, EAGeR Trial, 2007-2011^{a,b}

	N ^c	Among complete follow-up (n=1,094) Adjusted ^f RR (95% CI)	Among pregnancies (n=803) ^c Adjusted ^f RR (95% CI)	Among live births (n=601) ^d Adjusted ^f RR (95% CI)
Original stratum				
<30 ng/mL (referent)	60	1	1	1
≥30 ng/mL	89	1.48 (1.12, 1.96)	1.29 (0.98, 1.69)	1.18 (0.91, 1.53)
Expanded stratum				
<30 ng/mL (referent)	90	1	1	1
≥30 ng/mL	81	1.06 (0.79, 1.41)	0.99 (0.73, 1.33)	1.01 (0.74, 1.42)

^a8 twin gestations contributed two observations each to the analysis.

^bRRs and 95% CIs calculated using multiply-imputed generalized estimating equations of log-binomial regression with robust standard errors and stabilized inverse-probability weights to account for loss to follow-up. Poisson models used in instances of model non-convergence.

^cModels further weighted to account for selection of pregnancies.

^dModels further weighted to account for selection of pregnancies and survival to live birth.

^eNumber of live-born males.

^fMultivariable model adjusted for age (continuous), race/ethnicity (white or non-white), number of previous live births (0, 1, or ≥2).

Also, I'm not sure their last statement in their quote above is correct. I haven't seen the data supporting it, but neither have I seen data suggesting it is incorrect. I think that women with losses are now encouraged to go ahead and try again as soon as they're ready, while in the past I think there were some who

encouraged “taking a rest” before trying again. Regardless, they expanded the recruitment presumably because it was taking so long to enroll with just the original enrollment criteria.

Response: We thank the Reviewer for this opportunity to clarify. We agree that there aren’t strong data to support or refute the above statement and would like to note that the above quotation appears only in a previous response letter (not in the manuscript text). Please note the following changes to the manuscript that address differences in estimates by eligibility criteria:

Lines 122-126: “In post-hoc analyses stratified by eligibility criteria, estimates describing the association of vitamin D status and male live birth were pronounced in the original stratum among women with a single recent loss (RR = 1.48; 95% CI = 1.12, 1.96) compared to the women in the expanded stratum with 1 or 2 prior losses at any time in the past and at any gestational age (RR= 1.06; 95% CI = 0.79, 1.41)”

Lines 180-183: “Moreover, associations were particularly observed among women with a single recent loss, who tended to have lower adiposity on average, which may suggest that the influence of vitamin D on male live birth may be stronger among women with non-obesity-related inflammation.”

The Reviewer is correct that recruitment for a trial among women with a single recent loss (<20 weeks gestational age during the last year) is challenging. As a precaution, the trial was designed to allow for a second randomization stratum from the outset. After 2 months of recruitment and enrolling only nine women, eligibility was expanded to allow for women with 1-2 losses at any gestational age and at any time in the past. In addition to the practical aspects of this recruitment strategy, this also enabled us to target two important populations. The ‘original’ stratum could be thought of as the ‘biologically-based inclusion criteria’. It is hypothesized that women with a single recent loss might have an endometrium that is in the healing process and may be more likely to benefit from the positive effect of low dose aspirin. However, knowing that recruitment for this specific population would be challenging, and that findings from this trial would likely be applied in practice with small deviations from the original inclusion criteria (i.e., “indication creep”), we also enrolled women in the ‘expanded’ stratum. This ‘expanded’ population is what we would consider the more pragmatic trial as these inclusion criteria are more open and akin to how the intervention would be applied in practice. These issues are also discussed in detail in a prior paper. (Mumford et al 2019) In thinking about the stratified results presented above, it remains that these results would tend to suggest that the influence of vitamin D on male live birth may be stronger among women with non-obesity related inflammation. Moreover, given the hypotheses of the overall trial, it may also suggest that women with a single recent loss who have an endometrium in the healing process may be more amenable to the anti-inflammatory effects of vitamin D. As discussed in our response to the Reviewer’s comment above, we now discuss the stratified findings in the revised manuscript. We are careful in the interpretation of the findings given the analysis was performed post-hoc.

Reference: Mumford, S.L., Schisterman, E.F. New methods for generalizability and transportability: the new norm. *Eur J Epidemiol* 34, 723–724 (2019).
<https://doi.org/10.1007/s10654-019-00532-3>

2. I’m still unclear about the value of the karyotype data. They note in their response that “The goal of

incorporating the karyotype data was to estimate the association of preconception vitamin D and sex ratio shortly after implantation.” Of course, that is not possible with karyotype data.

Response: In response to this suggestion, we removed all results that incorporated karyotype data from the main tables and placed them in Supplemental Material (now appear as Supplemental Tables 3 and 4). We also made the following revisions to the text to indicate that these were exploratory analyses. We feel that this novel, though limited, aspect of our data is an important and interesting contribution that we prefer to keep in the paper but are happy to remove it at the editor’s request.

Lines 103-106: “In exploratory analyses that evaluated the probability of pregnancy with a male, women with sufficient preconception vitamin D levels were more likely to have a pregnancy with a male fetus compared to those with insufficient levels (36% sufficient and 29% insufficient; adjusted RR = 1.22, 95% CI = 1.01, 1.47) (**Supplemental Table 3**).”

Lines 377-379: “In exploratory analyses, we evaluated associations of vitamin D status and pregnancy with a male fetus utilizing the available karyotype data for 56 out of the 135 clinical pregnancy losses (including 2 twin gestations).

3. Sex-ratio differences rarely replicate. Given the lack of consistency of results between recruitment subsets in this cohort, and given that the only statistically-significant findings in Table 2 are those that rely on IPW weightings to include women who did not conceive, perhaps it would be best to present the results as hypothesis-generating findings.

Response: First, we hope that we have sufficiently addressed the Reviewer’s concerns regarding effect modification by eligibility criteria by adding the results of these analyses to Supplemental Material and a corresponding discussion in the text (see above).

Second, regarding the Reviewer’s point about statistical significance, we would like to highlight that the point estimates from each of these three analyses are quite consistent, with expected losses of efficiency (wider confidence intervals) due to smaller sample sizes in analyses restricted to pregnancies and live births. To interpret the estimates among all women who completed follow-up (RR = 1.26; 95% CI = 1.03, 1.56) as evidence of an association, but the estimates among pregnancies (RR = 1.18; 95% CI = 0.97, 1.44) and live births (RR = 1.18; 95% CI = 0.94, 1.48) as “null” or “no association” simply because of reduced efficiency would be incorrect. The pitfalls of such overreliance on significance testing have been well-described across scientific disciplines. (Amrhein et al, 2019)

Third, we would like to clarify that, the analyses (among 1,094 women) include a weight that only addresses loss to follow-up and does not “rely on IPW weightings to include women who did not conceive,” as the Reviewer suggests.

Finally, as recommended by the Reviewer, we have added “hypothesis-generating” language in the conclusions on lines 257-259: “Replication of these hypothesis-generating findings, as well as evaluations of other dietary and lifestyle factors that may influence sex ratio, are warranted in future investigations.”

Reference: Amrhein V, Greenland S, McShane B. Scientists rise up against statistical significance. *Nature*. 2019 Mar;567(7748):305-307. doi: 10.1038/d41586-019-00857-9. PMID: 30894741.

REVIEWERS' COMMENTS

Reviewer #5 (Remarks to the Author):

The authors provide a comprehensive response to the concerns raised by Reviewer 2. The study design is sound, albeit not perfect. But secondary analysis of the EaGER trial data to contribute to the literature on sex ratios over the course of gestation in relation to vitamin D levels seems entirely appropriate. The analysis is thorough and thoughtful. A major strength of the paper is the delineation of three separate research questions, with modified denominators to reflect persons at risk. The explanations for their approach to addressing the three research questions is quite clear and well founded.

The authors also employ sensitivity analyses to assess some sources of bias. They added an additional one in their response document, which shows a stronger association among women with shorter intervals between blood draw and conception, and adds further support to the authors' findings and interpretation. They also address other concerns related to loss to follow-up bias. They control for several potential confounders, including season and supplementation. The overall analyses, sub-analyses, and bias analyses reflect rigorous epidemiology.

The authors' interpretation of their findings is not overstated, in my opinion. The difference in magnitude of effects for the two recruitment strata is curious, and it is important that the authors include those results. While there can be no known explanation for the difference in associations, the authors are appropriately careful in their interpretation while also offering an interesting hypothesis.

REVIEWERS' COMMENTS

Reviewer #5 (Remarks to the Author):

The authors provide a comprehensive response to the concerns raised by Reviewer 2. The study design is sound, albeit not perfect. But secondary analysis of the EaGER trial data to contribute to the literature on sex ratios over the course of gestation in relation to vitamin D levels seems entirely appropriate. The analysis is thorough and thoughtful. A major strength of the paper is the delineation of three separate research questions, with modified denominators to reflect persons at risk. The explanations for their approach to addressing the three research questions is quite clear and well founded.

The authors also employ sensitivity analyses to assess some sources of bias. They added an additional one in their response document, which shows a stronger association among women with shorter intervals between blood draw and conception, and adds further support to the authors' findings and interpretation. They also address other concerns related to loss to follow-up bias. They control for several potential confounders, including season and supplementation. The overall analyses, sub-analyses, and bias analyses reflect rigorous epidemiology.

The authors' interpretation of their findings is not overstated, in my opinion. The difference in magnitude of effects for the two recruitment strata is curious, and it is important that the authors include those results. While there can be no known explanation for the difference in associations, the authors are appropriately careful in their interpretation while also offering an interesting hypothesis.

Response: Thank you very much for your thoughtful review of our paper.